# Microbiota regulates the TET1-mediated DNA hydroxymethylation program in innate lymphoid cell differentiation

Xusheng Zhang[1,2,4], Xintong Gao[1,2,4], Zhen Liu[1,2,4], Fei Shao[1,2], Dou Yu[1,2], Min Zhao[1], Xiwen Qin[3] & Shuo Wang [1,2] ✉

Innate lymphoid cell precursors (ILCPs) develop into distinct subsets of innate lymphoid cells (ILCs) with specific functions. The epigenetic program underlying the differentiation of ILCPs into ILC subsets remains poorly understood. Here, we reveal the genome-wide distribution and dynamics of the DNA methylation and hydroxymethylation in ILC subsets and their respective precursors. Additionally, we find that the DNA hydroxymethyltransferase TET1 suppresses ILC1 but not ILC2 or ILC3 differentiation. TET1 deficiency promotes ILC1 differentiation by inhibiting TGF-β signaling. Throughout ILCP differentiation at postnatal stage, gut microbiota contributes to the downregulation of TET1 level. Microbiota decreases the level of cholic acid in the gut, impairs TET1 expression and suppresses DNA hydroxymethylation, ultimately resulting in an expansion of ILC1s. In adult mice, TET1 suppresses the hyperactivation of ILC1s to maintain intestinal homeostasis. Our findings provide insights into the microbiota-mediated epigenetic programming of ILCs, which links microbiota-DNA methylation crosstalk to ILC differentiation.

Innate lymphoid cells (ILCs) are innate lymphocytes that lack adaptive antigen receptors and exist primarily in mucosal surfaces, where they play important roles in enhancing resistance against pathogens, facilitating tissue repair, and regulating metabolism[1–3]. The differentiation and function of ILC subsets rely on lineage-specific transcription factors and cytokines. Common ILC precursors (ILCPs) are a subset of cells that highly express Id2 and PLZF and give rise to the formation of ILC1, ILC2, and ILC3 subsets[4,5]. During ILC differentiation, the differential expression of T-bet, GATA3 and RORγt affects the fates and functions of ILC precursors[6–8]. However, the epigenetic regulation of cell fate decisions in homogeneous ILC progenitor cell populations has not been well studied.

DNA methylation has been shown to play an important role during hematopoiesis by contributing to the formation of stable and heritable gene expression patterns. As a previous study reported, highly expressed genes are sparsely methylated in promoter regions, whereas silenced, nontranscribed genes show high levels of cytosine methylation in their promoters[9,10]. In addition, oxidation forms of 5-methylcytosine (5mC) of genomic DNA are an important epigenetic modification that influences gene expression[11,12]. Among the three oxidation forms of 5mC, 5-hydroxymethylcytosine (5hmC) is the most stable and abundant. Recently, an increasing number of studies have suggested that 5hmC is enriched around the transcription factor-binding sites, enhancers, gene bodies and promoters of highly expressed genes[13–16]. During DNA demethylation, 5hmC acts as a key intermediator in active demethylation pathways. Ten-eleven translocation (TET) family enzymes are DNA hydroxymethyltransferase and facilitate DNA hydroxymethylation[17,18]. Deficiency of TET is closely associated with developmental defects, immune cell dysfunction and malignancy[19–21]. However, the regulation of TET during ILC differentiation is barely known.

The microbiota on mucosal surfaces plays a pivotal role in the regulation of the host's immune system. At the early life stage,

[1]CAS Key Laboratory of Pathogen Microbiology and Immunology, Institute of Microbiology, Chinese Academy of Sciences, 100101 Beijing, China. [2]University of Chinese Academy of Sciences, 100049 Beijing, China. [3]Division of Infectious Diseases, Department of Medicine, Washington University School of Medicine, Saint Louis, MO, USA. [4]These authors contributed equally: Xusheng Zhang, Xintong Gao, Zhen Liu. ✉e-mail: wangshuo@im.ac.cn

colonization of commensal bacteria is critical for the education of host immunity[22–24]. Studies on germ-free (GF) animals have demonstrated that the absence of commensal microbes is associated with profound defects of intestinal immune functions[23,25]. Th17 cells are not observed in GF mice and are inducible upon commensal bacteria colonization[26,27]. Group 3 ILCs (ILC3s) are regulated by microbial metabolites or control the colonization of microbiota[28,29]. Recent studies have found that the intestinal microbiota is a vital factor in the regulation of ILC differentiation[30]. However, whether there is cross-talk among epigenetic regulation, microbiota and the metabolites during ILC development remains elusive.

In this study, we performed genome-wide characterization of 5mC and 5hmC in genome of various ILC subsets and their precursor cells. We found that TET1 and TET1-mediated regulation of TGF-β signaling suppressed ILC1 differentiation from ILCPs. During the early lifespan, the gut microbiota decreased the level of cholic acid, resulting in the down-regulation of TET1 expression and expansion of ILC1s. In adult life, TET1 controls the differentiation and activation of ILC1s to maintain intestinal homeostasis. In summary, these findings indicate that the microbiota modulates the DNA methylation program to orchestrate ILC1 differentiation and intestinal homeostasis.

## Results

### Genome-wide 5hmC and 5mC distribution in ILC and ILCP subsets

To analyze the DNA methylation profiles in ILC subsets and their respective precursors, we isolated ILC precursors (including ILCPs, ILC1Ps, ILC2Ps and ILC3Ps), and ILC subsets (including ILC1s, ILC2s and ILC3s) for methylated DNA immunoprecipitation sequencing (MeDIP–seq) and hydroxymethylated DNA immunoprecipitation sequencing (hMeDIP–seq), which positively enrich methylated and hydroxymethylated DNA, respectively[31]. The distribution of 5hmC and 5mC within gene bodies and their regulatory elements (including promoters and CpG islands (CpGIs)) were analyzed. All ILC subsets displayed a decrease of 5hmC and 5mC levels at transcription start sites (TSS) and transcription end sites (TES) (Fig. 1a). Similarly, the levels of 5hmC and 5mC were also decreased in the promoter regions of ILC1s, ILC2s and ILC3s as well as their respective precursors (Fig. 1b). Notably, we found that ILCPs displayed high levels of 5hmC and 5mC in the gene bodies and promoter regions (Fig. 1a, b), suggesting a DNA demethylation program from ILCPs to ILC subsets. In mammals, the CpGs are unmethylated in CpGI regions[31]. We next analyzed the distribution of 5hmC and 5mC in CpGI regions and found that 5hmC and 5mC levels were significantly downregulated at the centers of CpGIs in all ILC subsets (Fig. 1c). Similarly, ILCPs have the highest levels of 5hmC and 5mC in CpGIs, suggesting that the DNA demethylation around TSSs and CpGIs might play a critical role during ILC subsets differentiation.

Next, we analyzed the distribution of 5hmC and 5mC in regulatory elements (including the promoter, intron, and intergenic regions) of ILC subsets (Fig. 1d). Similar levels of hydroxymethylation and methylation of intron and intergenic regions were observed in different ILC subsets. However, the greatest changes of DNA hydroxymethylation and methylation were observed in the promoter regions during ILC differentiation (Fig. 1d). During ILC differentiation from their respective precursors, the methylation and hydroxymethylation of genes related to carbohydrate metabolism, lipid metabolism, or amino acid metabolism decreased, indicating metabolism modulation during ILC differentiation (Supplementary Fig. 1a–f). We analyzed the differentially hydroxymethylated promoters (DHMPs) or differentially methylated promoters (DMPs) during the differentiation of ILC from their precursors. Notably, there were more downregulated 5hmC and 5mC regions than upregulated regions in gene promoters (Fig. 1e). Moreover, the promoters of the genes related to immune cell activation, proliferation, and differentiation were

hydroxymethylated in ILC1s, ILC2s, and ILC3s, indicating that immune effector genes were epigenetically regulated in these cells (Fig. 1f).

We next analyzed the correlation of DHMPs or DMPs with differentially expressed genes (DEGs) between various ILC subsets (Supplementary Fig. 1g, h). Hypomethylation of gene promoters mainly contributed to the differential gene expression pattern in ILC subsets (Supplementary Fig. 1g, h). We further analyzed the 5hmC and 5mC distribution of lineage-specific genes of ILC subsets. The lineage-specific genes were divided into 12 clusters according to their 5hmC and 5mC distribution patterns in promoter regions (Fig. 1g). Signature cytokine genes of ILC subsets showed high levels of 5hmC or low levels of 5mC in promoter-proximal regions of the corresponding ILCs, including *Ifng* and *Tnf* for ILC1s, *Il5* and *Il4* for ILC2s, and *Il17f* for ILC3s (Supplementary Fig. 1i). Furthermore, we analyzed the 5hmC and 5mC distribution of lineage-specific genes in ILC subsets. Lineage-specific transcription factors (i.e., T-bet, GATA3, and RORγt) are key fate-determining factors during the differentiation of ILCs[2]. The promoters of the *Tbx21*, *Gata3* and *Rorc* genes display low levels of 5mC in ILC1s, ILC2s, and ILC3s respectively, as well as in their respective precursors (Fig. 1h and Supplementary Fig. 1j). Intriguingly, 5mC peaks were also enriched in upstream regions of lineage-specific transcription genes (i.e., *Tbx21*, *Gata3* and *Rorc*) of ILCPs (Supplementary Fig. 1j), suggesting an underlying epigenetic regulatory mechanism from ILCPs to ILC subsets. Our results revealed that DNA hydroxymethylation/demethylation, especially promoter hydroxymethylation/demethylation, might play a pivotal role during ILC differentiation.

### TET1 suppresses ILC1 differentiation from ILCPs

Ten-eleven translocation (TET) family enzymes facilitate DNA hydroxymethylation and demethylation[32]. We next examined the expression levels of *Tet1*, *Tet2,* and *Tet3* in ILC and ILC precursor subsets (Supplementary Fig. 2a). Notably, the expression level of *Tet1* was high in ILCPs and decreased in ILC1Ps and ILC1s, implying a potential regulatory function of TET1 in ILC differentiation.

To further assess the role of TET1 in ILC differentiation, we generated a mouse model with conditional deletion of *Tet1* gene in ILC precursors. After abrogation of *Tet1* in ILCPs, the cell number of ILC1s but not ILC2s or ILC3s was increased in the intestine (Fig. 2a, b). The cell number of ILC1Ps was also elevated (Fig. 2c), suggesting that increase of ILC1Ps might contribute to the expansion of ILC1 population. However, the proportion of ILC1s in the lung and liver, and other ILC subsets was not affected in *Tet1^flox/flox^;Zbtb16-Cre* mice (Supplementary Fig. 2b–d). Moreover, we found that ILCPs and common lymphoid progenitors (CLPs) in bone marrow (BM) did not show apparent changes after depletion of *Tet1* (Supplementary Fig. 2e, f). To validate the differentiation potential of ILCPs with *Tet1* depletion, we isolated ILCPs from the BM of *Tet1^flox/flox^;Zbtb16-Cre* or *Zbtb16-Cre* mice and transferred them into immune-deficient mice. After 6–8 weeks of reconstitution, we found that *Tet1^-/-^* ILCPs gave rise to ILC1s with higher efficiency than control ILCPs (Fig. 2d), suggesting that TET1 suppressed the differentiation of ILCPs to ILC1s. Similar results were obtained from the in vitro differentiation assay. *Tet1^-/-^* ILCPs preferred to differentiate into ILC1s (Fig. 2e). We next investigated the effector function of ILC1s. Accompanying the increased ILC1 population, IFN-γ-positive ILC1s were also elevated in *Tet1^flox/flox^;Zbtb16-Cre* mice (Fig. 2f), indicating the enhanced inflammation associated with ILC1s. In summary, these results indicated that TET1 suppresses ILC1 differentiation from ILCPs. Furthermore, we analyzed the transcriptome data of human ILCs and ILCPs (Supplementary Fig. 3a, b). Intriguingly, ILCPs with low expression of *TET1* were closer to ILC1s, especially intestinal ILC1s (Supplementary Fig. 3c–f), indicating that TET1 was related to the suppression of ILC1 differentiation from ILCPs in human intestine tissues.

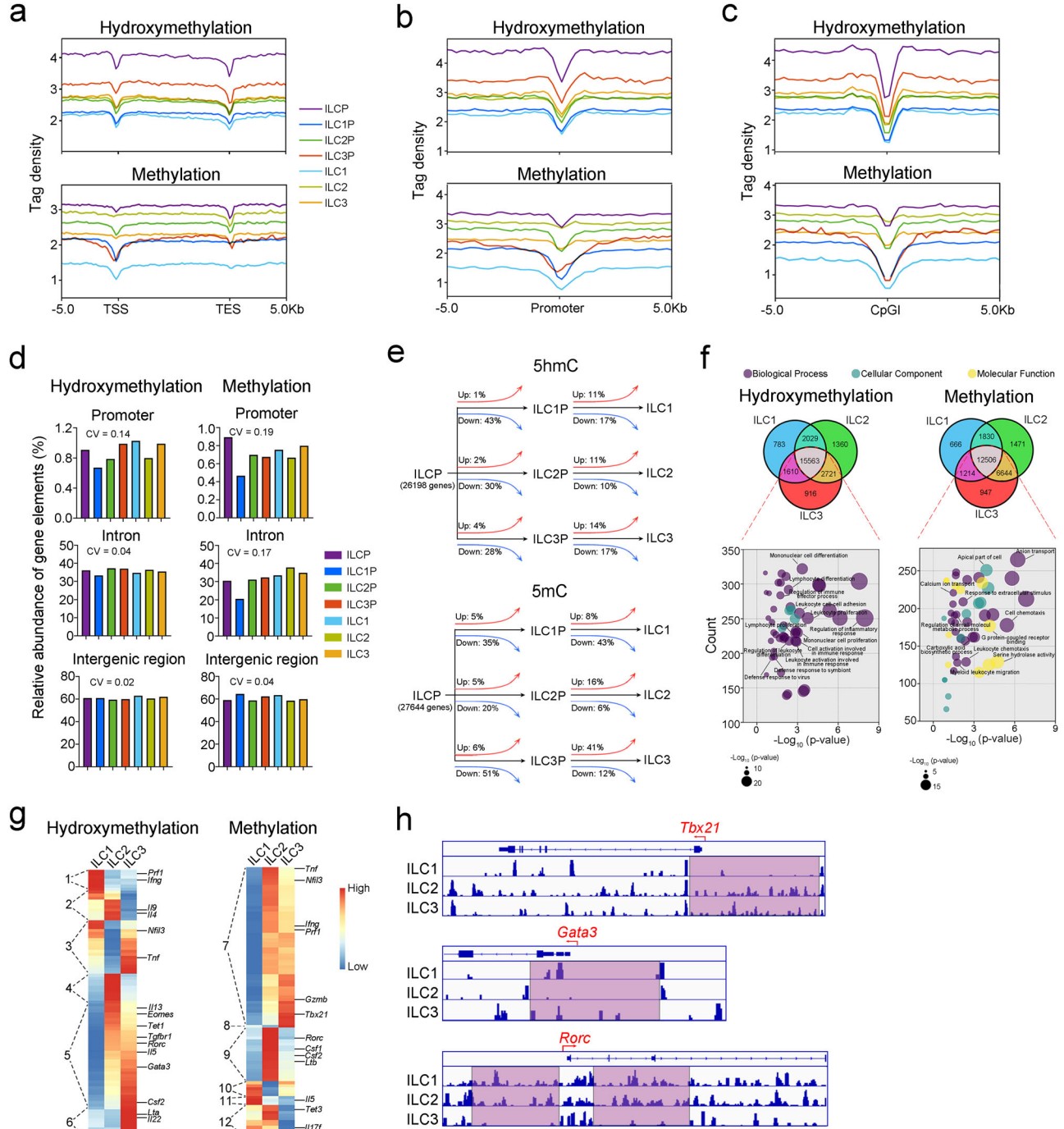

## TET1 promotes hydroxymethylation of the *Tgfbr1* promoter and inhibits ILC1 differentiation

To explore the regulation of ILC differentiation mediated by TET1, we analyzed the 5hmC DNA modification and transcription of genes in control (*Tet1*[+/+] ILCPs) and *Tet1*[-/-] ILCPs (Fig. 3a and Supplementary Fig. 3g). The 5hmC level of the genome in ILCPs was significantly decreased after abrogation of TET1 (Fig. 3a). After comparison with the gene expression pattern, we found that transcriptome changes were correlated with DNA hydroxymethylation modification along with TET1 abrogation (Fig. 3b, c). Gene set enrichment analysis (GSEA) of differentially hydroxymethylated genes overlapping with DEGs showed that the gene set of TGF-β signaling pathway was specifically enriched in TET1-completed ILCPs (Fig. 3d). The genomic distribution of down-regulated 5hmC level and up-regulated 5mC level at the loci of the *Tgfbr1* gene, but not *Tgfbr2* and *Tgfbr3*, was apparently observed

after depletion of *Tet1* in ILCPs (Fig. 3e–g), indicating that TET1 modulated the methylation of the *Tgfbr1* gene. We further found that TET1 was able to bind TSS upstream regions of the *Tgfbr1* in ILC1Ps and ILCPs (Fig. 3h). Gene expression analysis showed that the expression level of *Tgfbr1* decreased in *Tet1*[-/-] ILCPs (Fig. 3i). Next, we used in vitro culture to analyze the influence of TGF-β1 signaling on ILC1 differentiation. The addition of TGF-β1 inhibited the differentiation of ILC1s from ILCPs, which was abrogated by the TGF-βR1 inhibitor (Fig. 3j). TGF-βR1 inhibitor significantly promoted ILC1 differentiation (Fig. 3j). However, the effect of TGF-β1 was not observed in *Tet1*[-/-] ILCPs (Fig. 3k), suggesting that TET1 modulated ILC1 differentiation from ILCPs via a TGF-β1 signaling-dependent manner. Collectively, our findings showed that TET1 suppresses the differentiation of ILC1s by promoting *Tgfbr1* gene hydroxymethylation and demethylation, resulting in activation of TGF-β1 signaling.

**Fig. 1 | Genome-wide distribution of 5hmC and 5mC in ILCs and ILC precursors.**
**a** The normalized tag density profiles of 5hmC and 5mC distribution across gene body ± 5 kb flanking regions in ILC1s, ILC2s and ILC3s as well as their respective precursors[53]. ILC1s (ILC1 = Lin−CD45+CD127+NK1.1+NKp46+, Lin = CD3, CD19, CD11b, Gr1, Ter119, CD45R), ILC2s (ILC2 = Lin−CD45+CD127+KLRG1+, Lin = CD3, CD19, CD11b, Gr1, Ter119, CD45R, NK1.1) and ILC3s (ILC3 = Lin−CD45+CD127+RORγt-GFP+, Lin = CD3, CD19, CD11b, Gr1, Ter119, CD45R, NK1.1) from lamina propria of small intestine and ILC1 precursors (ILC1Ps) (ILC1Ps = Lin−CD45+CD127+NK1.1+NKp46+CD49a+, Lin = CD3, CD19, CD11b, Gr1, Ter119, CD45R), ILC2 precursors (ILC2Ps) (ILC2Ps = Lin−CD45+ST2+KLRG1−, Lin = CD3, CD19, CD11b, Gr1, Ter119, CD45R), ILC3 precursors (ILC3Ps) (ILC3Ps = Lin−CD45+CD127+α4β7intRORγt-GFP+, Lin = CD3, CD19, CD11b, Gr1, Ter119, CD45R) and common ILC precursors (ILCPs) (ILCPs = Lin−c-Kit+CD127+α4β7+PLZF-GFP+, Lin = CD3, CD19, CD11b, Gr1, Ter119, CD45R) from bone marrow (BM) were isolated from wild type (WT), RORγt-GFP and PLZF-GFP reporter mice followed by MeDIP-seq and hMeDIP-seq. Each ILC subset was collected from at least three mice. **b**, **c** The normalized tag density profiles of 5hmC and 5mC distribution around ± 5 kb regions flanking centers of promoter (**b**) and CpG island (CpGI) (**c**) regions of the genome in ILC1s, ILC2s, and ILC3s as well as their respective precursors. **d** The distribution of 5hmC and 5mC in regulatory elements (including promoter, intron, and intergenic regions) of ILC subsets. Frequencies of hydroxymethylated (left panel) and methylated (right panel) regulatory elements in each ILC subset were analyzed with MACS2 and Homer software. Peaks with q value (two-sided, Bonferroni-corrected p value) < 0.001 were selected and aligned to the mm10 reference genome for annotation. Coefficient of Variation (CV) of regulatory elements in ILC subsets was shown. **e** Dynamic changes in the 5hmC and 5mC distributions in promoter regions during ILC differentiation. A tag density of log₂(fold change) > 1 was considered as differentially hydroxymethylated promoters (DHMPs) or differentially methylated promoters (DMPs). The percentages of DHMPs and DMPs during ILC differentiation were analyzed. **f** Venn analysis of hypermethylated and hyperhydroxymethylated loci distribution in the gene promoters of ILC1s, ILC2s, and ILC3s. The peaks with a tag density over 3.8 (mean tag density of peaks) were considered hypermethylated or hyperhydroxymethylated peaks. The respective genes were selected for annotation with the GO database, p values were calculated by two-sided hypergeometric test. **g** The heatmap shows the methylation and hydroxymethylation levels of genes. The hypermethylated/hyperhydroxymethylated (red) and hypomethylated/hypohydroxymethylated (blue) promoters of the indicated genes in ILC1s, ILC2s, and ILC3s were classified into twelve clusters according to hydroxymethylation or methylation patterns. Gene promoters with high levels of hydroxymethylation in ILC1s, ILC1s/ILC2s, ILC1s/ILC3s, ILC2s, ILC2s/ILC3s, and ILC3s were classified into groups 1, 2, 3, 4, 5, and 6, respectively. Similarly, gene promoters with low levels of methylation in ILC1s, ILC1s/ILC2s, ILC1s/ILC3s, ILC2s, ILC2s/ILC3s, and ILC3s were classified into groups 7, 8, 9, 10, 11, and 12, respectively. **h** The distributions of 5mC across the indicated genes in each ILC subset were visualized by IGV. Source data are provided as a Source Data file.

## Gut commensal bacteria suppress TET1 expression and promote ILC1 differentiation

Previous reports have shown that ILCs undergo differentiation and expansion during the postnatal stage[33]. We found that ILC1s but not ILCPs underwent dramatic proliferation after weaning (at postnatal day 21) (Fig. 4a and Supplementary Fig. 4a). We also found that ILCPs existed in the intestine but not in other peripheral tissues, indicating that the differentiation of ILCPs in the gut might be affected by the microbiota (Supplementary Fig. 4b). Notably, the abundance of commensal bacteria apparently increased after weaning, accompanied with the expansion of ILC1s (Fig. 4b). Additionally, decreased expression levels of *Tet1* and *Tgfbr1* in the intestinal ILC1s and ILCPs were observed (Fig. 4c, d), indicating that the microbiota may promote the differentiation of ILC1s. The expression levels of *Tet2* and *Tet3* in ILC1s were not apparently changed from the postnatal stage to adulthood (Supplementary Fig. 4c, d). To verify the influence of the microbiota on ILC1 differentiation, we treated mice with antibiotics during the postnatal stage and found that the clearance of bacteria impaired the expansion of ILC1s (Fig. 4e), suggesting that the microbiota promotes ILC1 differentiation. ILCPs were not affected by microbiota (Supplementary Fig. 4e). In addition, the expression levels of *Tet1* and *Tgfbr1* in ILC1s or ILCPs were upregulated after antibiotic treatment (Fig. 4f, g). Moreover, the addition of a TGF-β1 receptor inhibitor apparently increased the ILC1 and ILC1P populations regardless of whether antibiotics were administered (Fig. 4h and Supplementary Fig. 4f), and ILCPs were not apparently changed (Supplementary Fig. 4g), indicating that the microbiota promotes the differentiation of ILC1s via suppression of TGF-β signaling.

During the postnatal stage, successful establishment of ILC1s plays a crucial role in resistance against pathogenic bacteria in adult mice. With antibiotic treatment at the postnatal stage, the body weight and gut bacteria decreased, accompanied by a reduced ILC1 population (Supplementary Fig. 5a–c). The populations of ILC1s in liver and lung (Supplementary Fig. 5d, e), as well as ILC2s, ILC3s, and ILCPs (Supplementary Fig. 5f–h) were not changed by antibiotic treatment. The mice with abrogation of commensal bacteria at weaning stage showed more susceptible to pathogen infection (Supplementary Fig. 5i). Taken together, microbiota promotes the differentiation of ILC1s and suppresses TET1 expression, which is beneficial for resistance to pathogen infection in adult mice.

## Intestinal microbiota prevents DNA hydroxymethylation and restores the ILC1 population

Next, we assessed the influence of the microbiota on ILC1 differentiation under germ-free conditions. The cell numbers of ILC1s and ILC1Ps, but not ILCPs were apparently decreased under germ-free conditions (Fig. 5a, b and Supplementary Fig. 5j). Notably, the ILC1 population could be restored after bacterial re-colonization of cohoused mice, indicating that gut microbiota contributed to the promotion of ILC1 differentiation. To understand the epigenetic modification of ILCPs by microbiota, we analyzed the DNA methylation and hydroxymethylation in ILCPs under germ-free conditions. Intriguingly, the levels of 5hmC throughout the gene bodies, CpG islands, and promoters significantly increased after ablation of the microbiota (Fig. 5c–e). Importantly, the level of 5hmC in ILCPs from cohoused mice was downregulated to the level of SPF mice (Fig. 5c–e), suggesting that the microbiota modulates the DNA hydroxymethylation program of ILCPs. We also analyzed the changes of 5hmC and 5mC levels of promoters in ILCPs after the transplantation of gut microbiota. We found that introducing microbiota significantly changed the levels of 5hmC and 5mC in promoters of ILCP genome (Fig. 5f–i). Importantly, the expression level of *Tet1* in ILCPs was decreased after establishment of the intestinal microbiota (Fig. 5j). Accordingly, we observed 5mC at loci of the *Tgfbr1* promoter accompanied by a decreased expression level of *Tgfbr1* in ILCPs (Fig. 5k, l). Therefore, the microbiota promotes ILCP differentiation by suppressing DNA hydroxymethylation and TGF-β1 signaling.

## Intestinal microbiota modulates intestinal metabolites and decreases TET1 level to promote ILC1 differentiation

According to our data, DNA methylation and ILC1 differentiation was affected by intestinal microbiota. Previous studies showed that gut commensal bacteria contribute to immune cells differentiation through metabolic changes[34]. To explore the potential mechanism of intestinal microbiota in changing the DNA methylation program and ILC1 differentiation, we collected small intestinal contents for metabolomics analysis. After antibiotic treatment, the metabolites in the intestine were significantly changed (Fig. 6a, b). The levels of bile acids were increased after antibiotic treatment, especially the primary bile acid: cholic acid. Notably, cholic acid is the most abundant bile acid in ABX-treated mice (Fig. 6c–e), suggesting that cholic acid might affect ILC1 differentiation in intestinal tract.

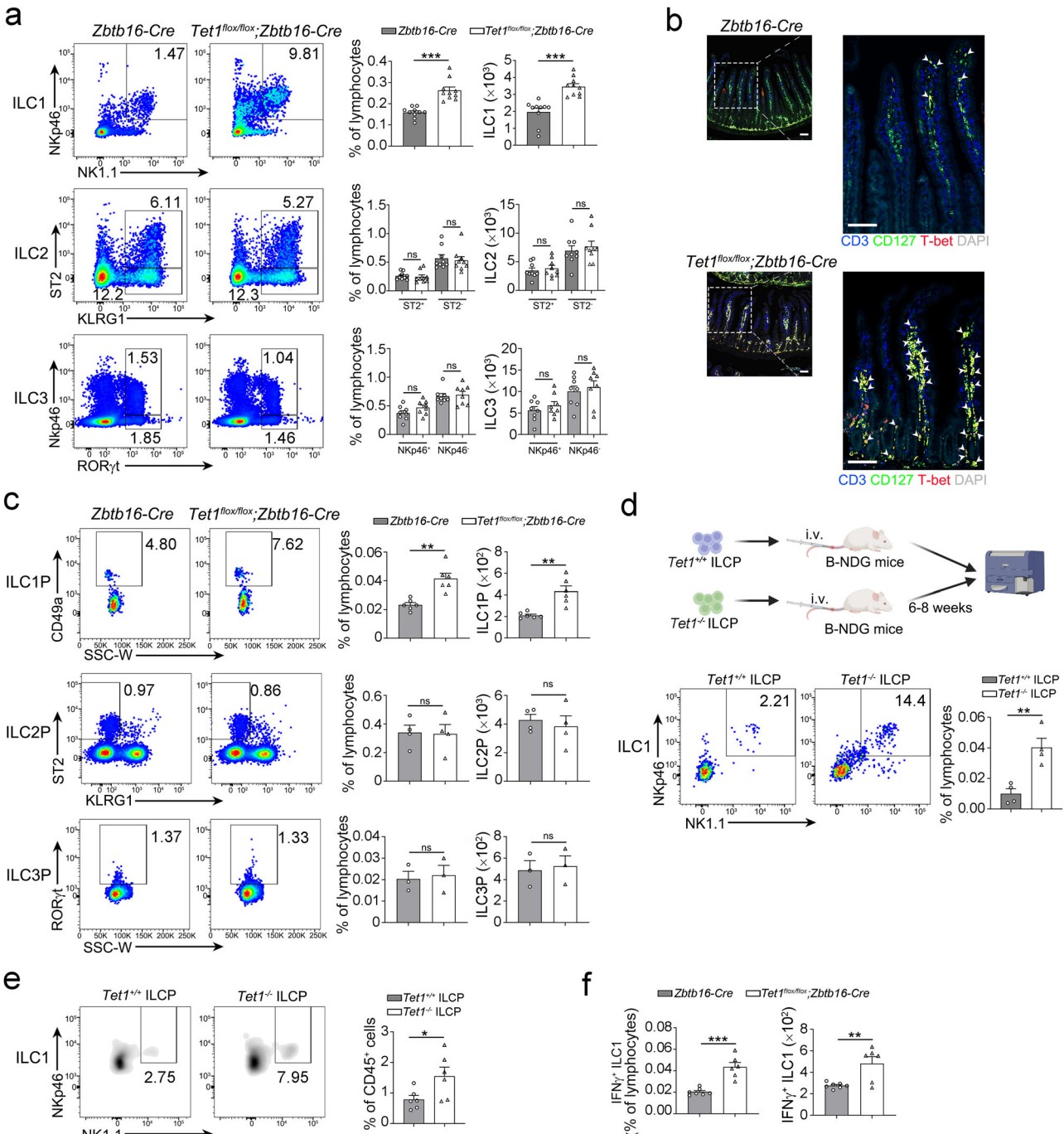

**Fig. 2 | TET1 inhibits the differentiation of intestinal ILC1s from ILCPs. a** Flow cytometry analysis of ILC1s, ILC2s, and ILC3s in the small intestine of *Zbtb16-Cre* mice and *Tet1^flox/flox^;Zbtb16-Cre* mice. The percentage and cell number of ILC1s, ILC2s, and ILC3s were shown as the mean ± SEM. ***p < 0.001; ns, not significant by two-sided unpaired Student's *t* test (*p* < 0.0001 for ILC1s; *p* = 0.69, 0.70, 0.58, 0.57 for ILC2s; *p* = 0.13, 0.76, 0.38, 0.61 for ILC3s). (n = 10 for ILC1s, n = 9 for ILC2s, n = 8 for ILC3s). **b** Immunofluorescence (IF) analysis of ILC1s in the small intestine of *Zbtb16-Cre* and *Tet1^flox/flox^;Zbtb16-Cre* mice. CD3, blue; CD127, green; T-bet, red; Nucleus, DAPI. Scale bar, 100 μm. The arrowheads indicated CD3⁻CD127⁺T-bet⁺ ILC1s. Data are representative of at least three independent experiments. **c** Analysis of ILC1Ps, ILC2Ps, and ILC3Ps of *Zbtb16-Cre* and *Tet1^flox/flox^;Zbtb16-Cre* mice by flow cytometry. The percentage and cell number of ILC1Ps, ILC2Ps, and ILC3Ps from *Zbtb16-Cre* and *Tet1^flox/flox^;Zbtb16-Cre* mice were shown as the mean ± SEM. **p < 0.01; ns, not significant by two-sided unpaired Student's *t* test (*p* = 0.0010, 0.0013 for ILC1Ps; *p* = 0.91, 0.61 for ILC2Ps; *p* = 0.79, 0.78 for ILC3Ps). (n = 6 for ILC1Ps, n = 4 for ILC2Ps, n = 3 for ILC3Ps). **d** Differentiation of *Tet1^+/+^* and *Tet1^−/−^* ILCPs in the intestine.

ILCPs from the BM of *Zbtb16-Cre* mice and *Tet1^flox/flox^;Zbtb16-Cre* mice were isolated and adoptively transferred into NOD.CB17-*Prkdc^scid^Il2rg^tm1^*/Bcgen (B-NDG) mice. After 6–8 weeks, the cell frequency of ILC1s in the gut was analyzed by flow cytometry and shown as the mean ± SEM (n = 4 independent biological replicates for each group). **p < 0.05 (*p* = 0.0043) by two-sided unpaired Student's *t* test. **e** TET1 suppressed ILC1 differentiation in vitro. ILCPs were isolated from the BM of *Zbtb16-Cre* mice and *Tet1^flox/flox^;Zbtb16-Cre* mice, and cultured in medium containing 200 ng/mL IL-7 and SCF on OP9-DL1 stromal cells for 7 days followed by flow cytometry analysis. The percentage and cell number of ILC1s were shown as the mean ± SEM. *p < 0.05 (*p* = 0.040) by two-sided unpaired Student's *t* test (n = 6 independent biological replicates for each group). **f** Expression of IFN-γ in intestinal ILC1s from the indicated mice was analyzed by flow cytometry. The percentage and cell number of IFN-γ⁺ ILC1s were shown as the mean ± SEM. **p < 0.01; ***p < 0.001 by two-sided unpaired Student's *t* test (*p* < 0.0001 and *p* = 0.0061). (n = 7 for *Zbtb16-Cre* mice, n = 6 for *Tet1^flox/flox^;Zbtb16-Cre* mice). Source data are provided as a Source Data file.

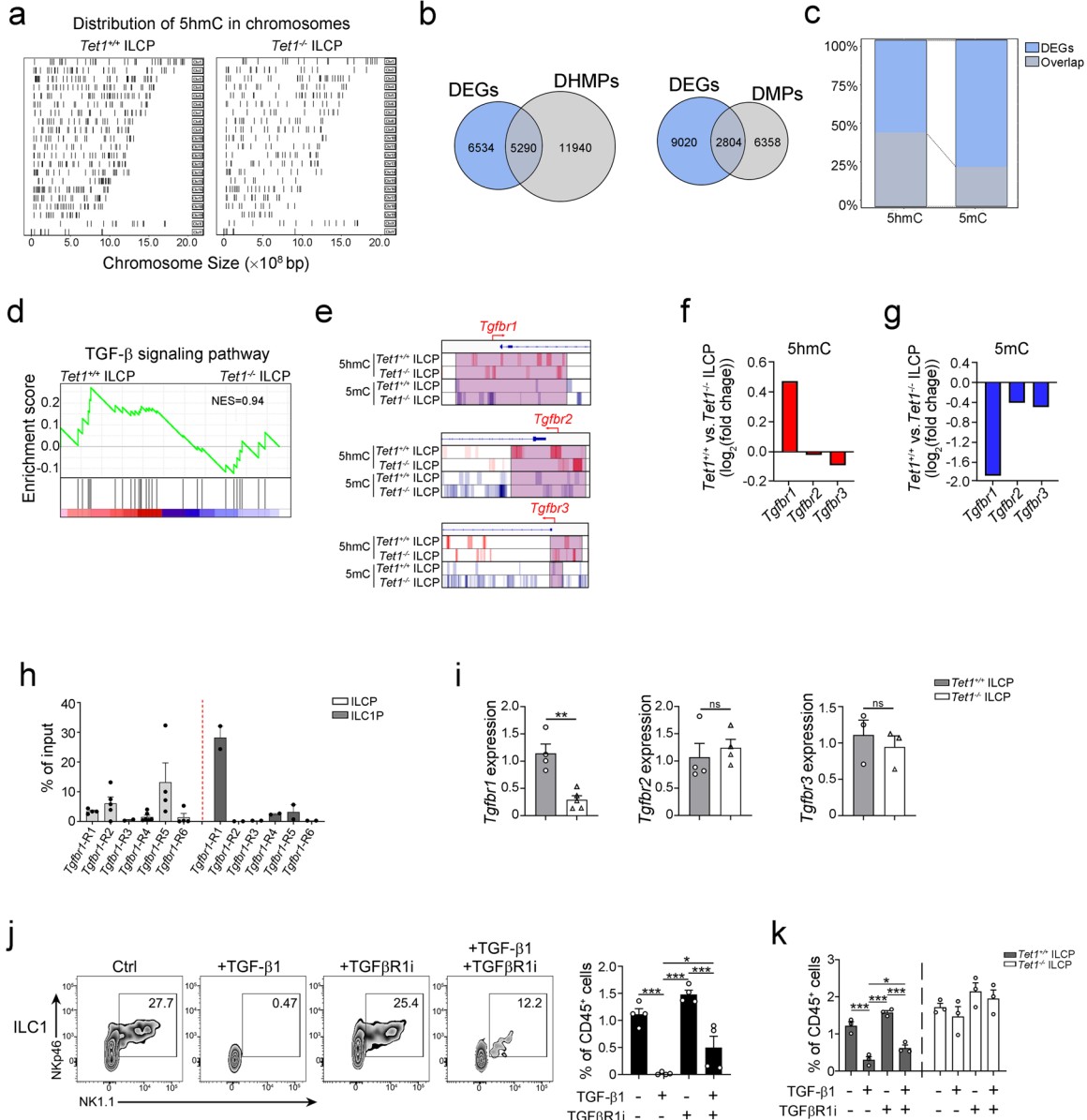

**Fig. 3 | TET1 contributes to the hydroxymethylation of the *Tgfbr1* gene and suppresses ILC1 differentiation. a** Genome-wide distribution of DNA hydroxymethylation in the genome of ILCPs from *Zbtb16-Cre* mice and *Tet1^flox/flox^;Zbtb16-Cre* mice. ILCPs were isolated from the bone marrow (BM) of the indicated mice for hMeDIP-seq. The DNA hydroxymethylation map was visualized with ChIPseeker and ClusterProfiler. **b** Venn analysis of DEGs and genes with DHMPs/DMPs in ILCPs from *Zbtb16-Cre* and *Tet1^flox/flox^;Zbtb16-Cre* mice. The overlapping regions showed the number of DEGs whose expression was correlated with DHMPs/DMPs. **c** The percentage of genes from the overlapping regions in (**b**) to all DEGs were shown. **d** Gene set enrichment analysis (GSEA) of DEGs that are correlated with DHMPs in ILCPs from *Zbtb16-Cre* mice comparing with *Tet1^flox/flox^;Zbtb16-Cre* mice. TGF-β signaling pathway gene set was enriched. **e** The distribution of 5hmC and 5mC across the *Tgfbr1, Tgfbr2* and *Tgfbr3* genes in ILCPs from *Zbtb16-Cre* mice and *Tet1^flox/flox^;Zbtb16-Cre* mice. The hydroxymethylated (red) and methylated (blue) loci in the *Tgfbr1, Tgfbr2, and Tgfbr3* genes were visualized by IGV. **f, g** Analysis of the hydroxymethylation and methylation levels of *Tgfbr1, Tgfbr2,* and *Tgfbr3* in ILCPs from *Zbtb16-Cre* mice and *Tet1^flox/flox^;Zbtb16-Cre* mice. The tag density of 5hmC (**f**) and 5mC (**g**) peaks around the TSSs of *Tgfbr1, Tgfbr2* and *Tgfbr3* were analyzed and shown. **h** TET1 binding sites on the *Tgfbr1* gene promoter in ILCPs and ILC1Ps were analyzed by ChIP-qPCR. ILCPs and ILC1Ps were isolated from the BM of *Zbtb16-GFP-Cre* mice followed by ChIP-qPCR. *Tgfbr1*-R1, R2, R3, R4, R5, and R6 respectively represented 100–300 bp, 300–600 bp, 600–800 bp, 800–1200 bp, 1200–1700 bp and 1700–2000bp upstream regions of the transcription start sites of *Tgfbr1* in

ILCPs and ILC1Ps. (ILCPs: n = 4 for *Tgfbr1*-R1, n = 5 for *Tgfbr1*-R2, n = 2 for *Tgfbr1*-R3, n = 6 for *Tgfbr1*-R4, n = 4 for *Tgfbr1*-R5, n = 4 for *Tgfbr1*-R6; ILC1Ps: n = 2 for each group). Data was shown as the mean ± SEM. **i** Analysis of *Tgfbr1, Tgfbr2* and *Tgfbr3* expression in ILCPs from *Zbtb16-Cre* mice and *Tet1^flox/flox^;Zbtb16-Cre* mice by qPCR. The relative expression of the indicated genes was shown as ± SEM. **p < 0.01; ns, not significant by two-sided unpaired Student's *t* test (*p* = 0.0016, 0.58, 0.55). (*Tgfbr1* expression: n = 4 for *Tet1^+/+^* ILCP, n = 5 for *Tet1^−/−^* ILCP; *Tgfbr2* expression: n = 4 for each group; *Tgfbr3* expression: n = 3 for each group). **j** TGF-β suppressed ILC1 differentiation from ILCPs in vitro. ILCPs were isolated from the BM and cultured in medium containing 200 ng/mL IL-7 and SCF on OP9-DL1 stromal cells in the presence or absence of 20 ng/mL TGF-β and/or 10 μM TGF-βR1 inhibitor (SB431542) for 7 days. PBS treatment served as control (Ctrl) group. The frequency of ILC1s was analyzed by flow cytometry and shown as the mean ± SEM. *p < 0.05; **p < 0.01; ***p < 0.001 by one-way ANOVA (*p* = 0.0002, <0.0001, 0.018, 0.0005). n = 4 for each group. **k** TET1 suppressed ILC1 differentiation via TGF-β signaling. ILCPs isolated from the BM of *Zbtb16-Cre* mice and *Tet1^flox/flox^;Zbtb16-Cre* mice were cultured in medium containing 200 ng/mL IL-7 and SCF on OP9-DL1 stromal cells in the presence or absence of TGF-β and/or TGF-βR1 inhibitor (SB431542) for 7 days. The frequency of ILC1s was analyzed by flow cytometry and shown as the mean ± SEM. *p < 0.05; **p < 0.01; ***p < 0.001 by one-way ANOVA (*p* = 0.0003, <0.0001, 0.029, 0.0002). n = 3 for each group. Source data are provided as a Source Data file.

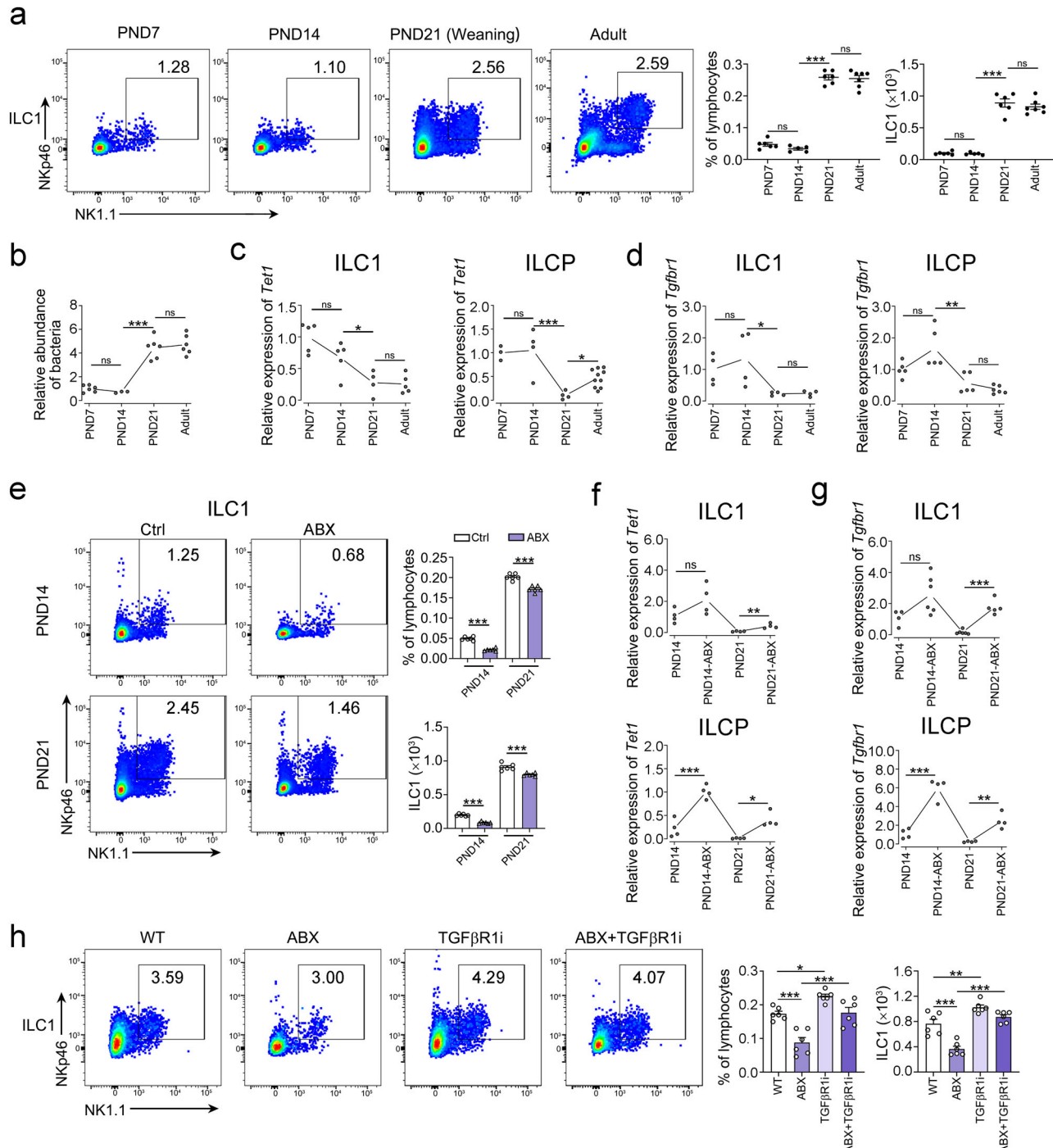

Next, we examined the effect of cholic acid on ILC1 differentiation. Intriguingly, cholic acid inhibited the differentiation of ILCPs to ILC1s in vitro (Fig. 6f). We further investigated the expression of cholic acid receptor-related genes in ILCPs. The expression level of *Gpbar1* gene (coding TGR5 protein) was high in ILCPs (Fig. 6g), suggesting that the TGR5 may play a crucial role in ILCPs differentiation. Additionally, TGR5 inhibitor rescue the differentiation of ILC1s from ILCPs which was suppressed by cholic acid, suggesting that cholic acid inhibited ILC1 differentiation dependently on TGR5 signaling (Fig. 6h). We also analyzed the expression of *Tet1* and *Tgfbr1* in ILCPs after the cholic acid treatment. The addition of cholic acid promoted the expression of *Tet1* and *Tgfbr1* in ILCPs, while the inhibition of TGR5 significantly decreased the *Tet1* and *Tgfbr1* expression level (Fig. 6i, j). We also analyzed the changes of 5hmC and 5mC levels of the *Tgfbr1* promoters

(Fig. 6k, l). We found that high level of 5hmC and low level of 5mC in the *Tgfbr1* promoter in ILCPs after cholic acid treatment, which can be rescued by TGR5 inhibitor (Fig. 6k, l). Collectively, the clearance of microbiota leads to the cholic acid accumulation in the intestine and inhibits ILC1 differentiation by promoting hydroxymethylation of *Tgfbr1* promoter.

### TET1 regulates ILC1 activation and maintains intestinal homeostasis at adult stage

According to our previous data, the gut microbiota modulates the DNA methylation profile via the downregulation of *Tet1* expression. Notably, the expression level of *Tet1* increased in ILCPs at the adult stage (Fig. 4c), and depletion of *Tet1* promoted the expansion and activation of ILC1s (Fig. 2a, b), indicating the regulatory role of TET1 in ILC1

**Fig. 4 | Gut commensal bacteria suppress *Tet1* expression and TGF-β signaling in ILC1s during the postnatal stage. a** Gut ILC1s from C57BL/6 mice at the indicated postnatal stages (from postnatal Day 7 (PND7) to adulthood, mice are weaned before PND21) were analyzed by flow cytometry. The percentage and cell number of ILC1s were analyzed by flow cytometry and shown as the mean ± SEM. ***$p < 0.001$; ns, not significant by one-way ANOVA ($p = 0.62$, <0.0001, 0.98, 0.99, <0.0001, 0.72). (n = 6 for PND7, n = 5 for PND14, n = 6 for PND21, n = 7 for Adult). **b** The relative abundance of intestinal bacteria in mice at the indicated postnatal stages were detected by qPCR. Mouse fecal DNA was extracted followed by qPCR analysis of bacteria 16S rRNA. The relative abundance of bacteria was shown as the mean ± SEM. ***$p < 0.001$; ns, not significant by one-way ANOVA ($p = 0.93$, <0.0001, 0.90) (PND7: n = 6 for PND7, n = 3 for PND14, n = 6 for PND21, n = 6 for Adult). **c, d** Expression levels of *Tet1* and *Tgfbr1* genes in ILC1s and ILCPs of mice at the indicated stage. ILC1s and ILCPs were isolated from intestine and bone marrow (BM) from WT mice, respectively. The gene expression of *Tet1* (**c**) and *Tgfbr1* (**d**) in ILC1s and ILCPs was analyzed by qPCR and shown as the mean ± SEM. *$p < 0.05$; **$p < 0.01$; ns, not significant by one-way ANOVA ($p = 0.11$, 0.016, 0.99, 0.79, <0.0001, 0.0259 for **c**, $p = 0.36$, 0.036, 0.99, 0.053, 0.0012, 0.80 for **d**) (*Tet1* expression in ILC1s: n = 5 for PND7, n = 5 for PND14, n = 4 for PND21, n = 5 for Adult; *Tet1* expression in ILCPs: n = 3 for PND7, n = 4 for PND14, n = 4 for PND21, n = 10 for Adult; *Tgfbr1* expression in ILC1s: n = 4 for each group; *Tgfbr1* expression in ILCPs:

n = 5 for PND7, n = 5 for PND14, n = 5 for PND21, n = 6 for Adult). **e** The ILC1s in the intestine of WT mice after ABX treatment were analyzed by flow cytometry. WT mice were treated with an antibiotic mixture (ABX) at PND7 for 14 days. The percentage and cell number of gut ILC1s from mice at the indicated stage were shown as the mean ± SEM. ***$p < 0.001$ by two-sided unpaired Student's *t* test. n = 6 for each group. **f, g** Expression levels of *Tet1* and *Tgfbr1* genes in ILC1s and ILCPs of mice after ABX treatment. ILC1s and ILCPs were isolated from the BM of mice before (PND14) or after weaning (PND21) with or without ABX treatment. The gene expression of *Tet1* (**f**) and *Tgfbr1* (**g**) in ILC1s and ILCPs was analyzed by qPCR and shown as the mean ± SEM. **$p < 0.01$; ***$p < 0.001$; ns, not significant by one-way ANOVA ($p = 0.075$, 0.0047, <0.0001, 0.017 for **f**, $p = 0.053$, <0.0001, <0.0001, 0.0064 for **g**) (*Tet1* expression in ILC1s and ILCPs: n = 4 for each group; *Tgfbr1* expression in ILC1s: n = 4 for PND14, n = 6 for PND14-ABX, n = 6 for PND21, n = 5 for PND21-ABX; *Tgfbr1* expression in ILCPs: n = 4 for each group). **h** Intestinal microbiota promoted the differentiation of ILC1s via TGF-β signaling. WT mice at PND7 were orally gavaged with ABX and intraperitoneally injected with 200 μL of 1 mg/mL TGF-βR1 inhibitor (TGFβR1i) twice a week for two weeks. The percentage and cell number of ILC1s were analyzed by flow cytometry and shown as the mean ± SEM. *$p < 0.05$; **$p < 0.01$; ***$p < 0.001$ by one-way ANOVA ($p = 0.0003$, 0.028, 0.0002, <0.0001, 0.0067, <0.0001). n = 6 for each group. Source data are provided as a Source Data file.

function at adult stage. Depletion of *Tet1* in ILCs showed apparent changes in the gut microbiota (Fig. 7a, b). Several bacterial genera, including *Prevotellaceae Ga6A1 group*, *Bacteroides* and *Lactobacillus*, were enriched in *Tet1^flox/flox^;Zbtb16-Cre* mice (Fig. 7a, b). Functional prediction of microbiota analysis revealed that the enriched bacteria were mainly pathogenic bacteria, which increased the risk of inflammatory disease (Fig. 7c). Accordingly, we observed aggravated intestinal inflammation in *Tet1^flox/flox^;Zbtb16-Cre* mice (Fig. 7d) and the enrichment of IBD-related genes (Fig. 7e) in ILC1s after depletion of *Tet1*. Moreover, the ILC1 population as well as IFN-γ⁺ ILC1s were significantly increased in the intestine of mice with DSS treatment in *Tet1^flox/flox^;Zbtb16-Cre* mice (Fig. 7f, g). The intestine showed more severe intestinal pathologies, suggesting that *Tet1* depletion in ILC1s enhanced susceptibility to intestinal inflammation (Fig. 7h).

To further investigate the effects of ILC1 epigenetic modifications on adult mice, we took advantage of a tamoxifen-induced depletion model. TET1 was abrogated in ILC precursors at the adult stage. In adult mice, the percentage and amount of ILC1s were apparently increased in *Tet1^flox/flox^;Id2-Cre* mice (Fig. 7i). Intestinal inflammation was aggravated after *Tet1* depletion in the adult mice (Fig. 7j). We also observed upregulated levels of IFN-γ expression in ILC1s (Fig. 7k) and enrichment of the IBD gene set (Fig. 7l) in *Tet1^-/-^* ILC1s. Furthermore, we analyzed the expression of *TET1* and TGF-β receptor genes in healthy people or IBD patients. IFNγ⁺ ILC1s were accumulated in the intestine of Crohn's disease patients with low expression levels of *TET1* and *TGFBR1* (Fig. 7m–o). Taken together, DNA methylation modification mediated by TET1 plays a crucial role in suppressing ILC1 hyper activation and maintaining intestine homeostasis.

## Discussion

During hematopoiesis, modification of the DNA methylation profile plays a critical role in establishing a specific functionality in each terminally differentiated subset. ILCs are endowed with different functions via epigenetic and transcriptional programs during differentiation from ILC precursors. In this study, we revealed the DNA methylation/hydroxymethylation profile during ILC differentiation. The DNA 5hmC modification of the *Tgfbr1* gene by TET1 augmented TGF-β signaling, which suppressed ILC1 development. Moreover, the gut microbiota decreased cholic acid level in the gut and suppresses *Tet1* expression and promotes ILC1 expansion during the postnatal stage. At the adult stage, TET1 contributes to the attenuation of ILC1 hyperactivation and maintenance of gut homeostasis.

DNA methylation is an epigenetic modification that can be stably inherited. During hematopoietic differentiation, DNA methylation patterns are changed to establish cell type-specific gene programs[34]. ILC precursors give rise to various ILC subsets accompanied by epigenetic modifications, including histone modification and regulation of chromatin accessibility[35,36]. However, the DNA methylation profile during ILC differentiation is unclear. Herein, we revealed the DNA methylation program from ILCPs to various ILC subsets and found that demethylation of the promoters of lineage-specific genes, including *Tbx21*, *Gata3*, and *Rorc*, is important for lineage specification. However, we also found that for some genes, DNA hydroxymethylation/demethylation of the promoter was not associated with gene expression. It is possible that DNA hyroxymethylation/demethylation in the gene body or enhancer might regulate the gene expression pattern. It is worth to define the lineage-specific DNA methylation/demethylation regions during ILC and other immune subset development.

Environmental factors play a critical role in the modification of DNA methylation. The heterogeneous niche with differential levels of cytokines and metabolites is closely associated with cell fate commitment[37,38]. At mucosal sites, immune cells are regulated by microbes[39]. There are large amounts of commensal bateria that interact with and affect the host immune system. Commensal bacteria show dynamic changes in the process of human growth[40], and are closely related to a variety of physiological and pathological processes[41–43]. The crosstalk between microbiota and epigenetic modification is not well defined. We showed that the gut microbiota contributes to the downregulation of *Tet1* gene expression and methylation of the *Tgfbr1* promoter, resulting in ILC1 expansion. According to our data, the cell number of ILC1s in other tissues, for example, lung and liver, was not significantly changed after TET1 depletion or clearance of bacteria (Supplementary Figs. 2b, 5d, e), indicating that gut microbiota contributes to the differentiation of ILC1s in the intestine but not in other tissues. Interestingly, we found ILCPs in the intestine but not in other peripheral tissues (Supplementary Fig. 4b). We also reveal that there are great differences in ILCs in different mucosal tissues[44]. It is possible that the gut microenvironment affects the differentiation of ILC1s from ILCPs in situ. Whether other factors contribute to the differentiation of ILC1s in other tissues is worthy of analysis. Moreover, we found that the level of TET1 is regulated by the gut microbiota. Microbiota degrade primary metabolites to secondary metabolites to modulate the intestine microenvironment[45]. We found that gut microbiota decreased cholic acid level and contributed to ILC1 differentiation. Cholic acid is reported associated with macrophage and Treg differentiation[46,47]. We

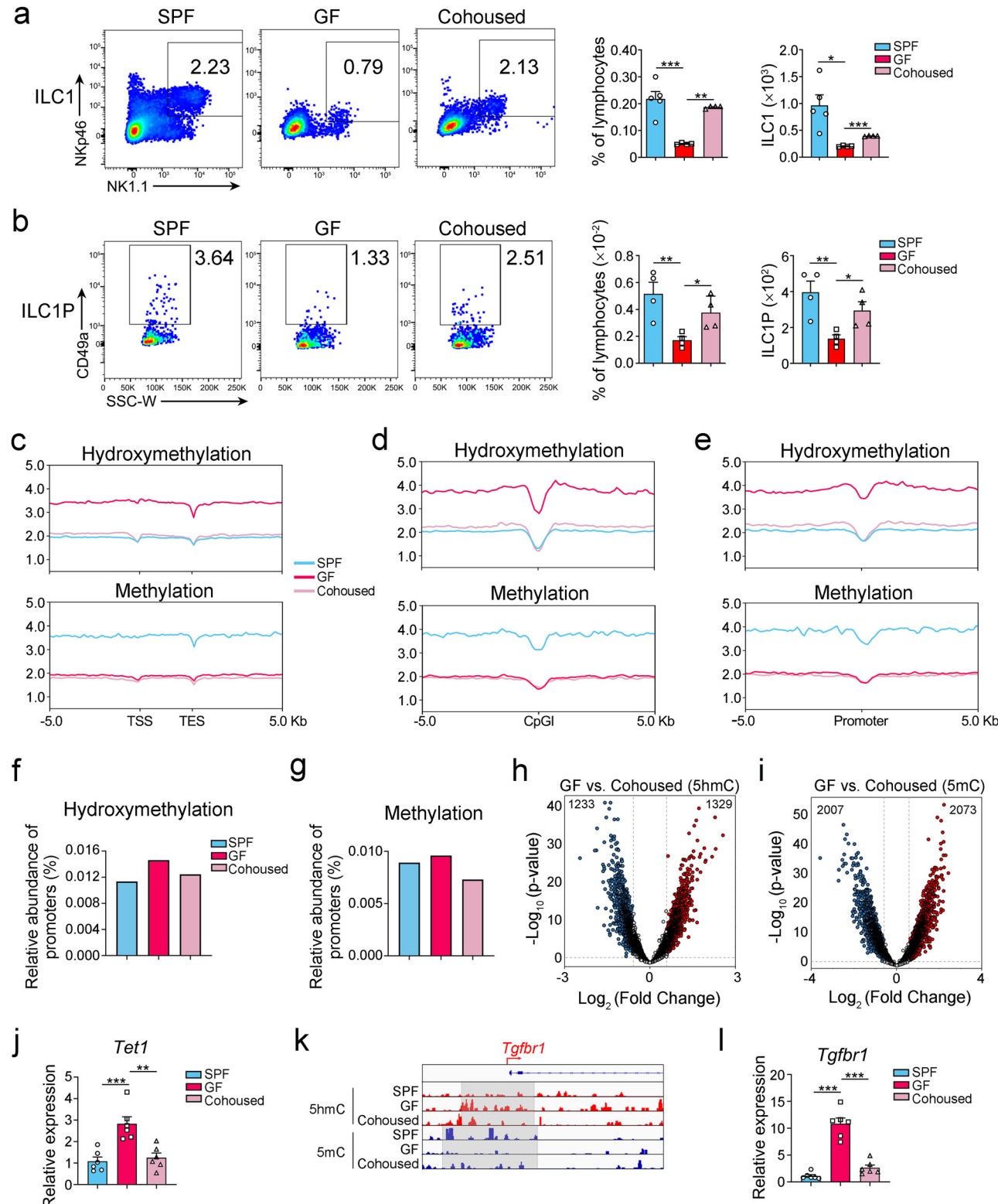

found that cholic acid suppresses ILC1 differentiation from ILCPs by increasing TET1 level. The regulatory mechanism of TET1 expression by cholic acid is worthy to be investigated. In addition to cholic acid, other metabolites from gut microbiota might regulate DNA methylation program of ILCs as well. Previous studies have showed that tryptophan enabled to modulate lymphocyte activation through indole-3-aldehyde (I3A)[48,49]. In our study, tryptophan and I3A in small intestinal content could not be detected at weaning stage. The role of tryptophan in regulating ILC1 differentiation is worthy to be studied. As

previously reported, segmented filamentous bacteria (SFB) initiate the priming of naïve T cells into Th17 cells[50]. The bacterial type that contributes to the epigenetic modification and differentiation of ILC1s needs to be defined. Moreover, the influence of the microbiota on the epigenetic regulation of other immune cells is also worthy of investigation.

In conclusion, the differentiation of the ILC subset is a complex process including DNA methylation modification along with microbiota and metabolite changes. Our study has revealed the pivotal role

**Fig. 5 | Gut microbiota modulates the DNA hydroxymethylation program from ILCPs to ILC1s. a, b** Gut microbiota affected the populations of ILC1s and ILC1Ps. WT mice under SPF, germ-free (GF) condition or GF condition followed by cohousing with SPF mice for two weeks (cohoused) were sacrificed for flow cytometry analysis of ILC1s and ILC1Ps. The percentage and cell number of ILC1s (**a**) and ILC1Ps (**b**) were analyzed by flow cytometry and shown as the mean ± SEM. *$p < 0.05$; **$p < 0.01$; ***$p < 0.001$ by one-way ANOVA ($p = 0.0002, 0.0013, 0.018, <0.0001$ for **a**, $p = 0.0038, 0.047, 0.0036, 0.042$ for **b**). (ILC1s: n = 5 for SPF, n = 4 for GF, n = 4 for Cohoused; ILC1Ps: n = 4 for each group). **c–e** Gut microbiota modulated DNA hydroxymethylation and methylation of genome in ILCPs. The normalized tag density profiles of 5hmC and 5mC distribution across gene body ± 5 kb flanking regions (**c**), and around center of CpG island (CpGI) (**d**) and promoter (**e**) in the genome of ILCPs were shown. ILCPs were isolated from WT C57BL/6 mice under SPF, GF or cohoused conditions and subjected to MeDIP-seq and hMeDIP-seq. The frequencies of hydroxymethylated (**f**) and methylated (**g**) promoters in ILCPs from the indicated mice were analyzed with MACS2 and Homer software. Peaks with q value (two-sided Bonferroni-corrected $p$ value) <0.001 were selected and aligned to

the mm10 reference genome for annotation. **h, i** Identification of DHMPs and DMPs of gene in ILCPs from germ-free mice compared with ILCPs from cohoused mice. The volcano plot shows the number of DHMPs (**h**) and DMPs (**i**) in each group. Blue dots represent the hyper-hydroxymethylated or hypermethylated promoters in the GF group. Red dots represent the hyper-hydroxymethylated or hypermethylated promoters in the cohoused group, $p$ values were calculated by two-sided unpaired Student's $t$ test. **j** Gut microbiota suppressed *Tet1* expression in ILCPs. ILCPs were isolated from mice under the indicated conditions and subjected to qPCR. The relative expression of *Tet1* was shown as the mean ± SEM. **$p < 0.01$; ***$p < 0.001$ by one-way ANOVA ($p = 0.0005, 0.0014$). n = 6 for each group. **k** The distribution of 5hmC and 5mC in *Tgfbr1* of ILCPs from SPF, GF, and cohoused groups. The hydroxymethylated (red) and methylated (blue) loci in *Tgfbr1* gene of ILCPs from indicated mice were shown. **l** Gut microbiota suppressed the expression of *Tgfbr1* in ILCPs. ILCPs were isolated from the indicated mice and subjected to qPCR. The relative expression of *Tgfbr1* was shown as the mean ± SEM. ***$p < 0.001$ by one-way ANOVA. n = 6 for each group. Source data are provided as a Source Data file.

of DNA methylation in ILC1 differentiation, in combination with TET1 loss and the function of the microbiota. Our work also established a rationale for targeting the crosstalk between the microbiota and epigenetic modifications to maintain intestinal homeostasis.

## Methods
### Study approval
All animal experiments were approved by the Institutional Ethics Committee of Institute of Microbiology, Chinese Academy of Sciences. The study is compliant with all relevant ethical regulations regarding animal research.

### Antibodies and reagents
Flow cytometry antibodies used in this study are as follows: anti-mouse CD3-eFluor 450 (17A2) (Cat# 48-0032-82, 1:500), anti-mouse CD19-eFluor 450 (1D3) (Cat# 48-0193-82, 1:500), anti-mouse NKp46-PE (29A1.4) (Cat# 12-3351-82, 1:500), anti-mouse KLRG1-APC (2F1) (Cat# 17-5893-82, 1:500), anti-mouse CD127-PerCP-eFluor 710 (SB/199) (Cat# 46-1273-82, 1:500), anti-mouse IL-33R-PE (RMST2-2) (Cat# 12-9333-82, 1:500), anti-mouse NK1.1-APC (PK136) (Cat# 17-5941-82, 1:500), anti-mouse PD1-PE/Cyanine7 (J43) (Cat# 25-9985-82, 1:500), anti-mouse lineage cocktail-eFluor 450 (17A2; RB6-8C5; RA3-6B2; Ter-119; M1/70) (Cat# 88-7772-72, 1:500) and anti-mouse RORγt-APC (AFKJS-9) (Cat# 17-6988-82, 1:300) were purchased from Invitrogen; anti-mouse CD49a-PE/Cyanine7 (HMα1) (Cat# 142607, 1:500), anti-mouse α4β7-APC (DATK32) (Cat# 120607, 1:500), anti-mouse c-Kit-PE (2BB) (Cat# 105807, 1:500), anti-mouse IFN-γ–APC/Cyanine7 (XMG1.2) (Cat# 505849, 1:500), anti-mouse Sca-1-FITC (W18174A) (Cat# 160907, 1:500) and anti-mouse CD45.2-FITC (30-F11) (Cat# 103107, 1:500) were from Biolegend; Rat IgG1 Isotype control (TNP6A7) (Cat# BP0290) was from BioXCell; anti-mouse TET1 (5D6) (Cat# 61941) was from Active Motif; anti-mouse IL-7R (G-11) (Cat# sc-514445, 1:500) and anti-mouse T-bet (4B10) (Cat# sc-21749, 1:500) were from SantaCruz; anti-mouse CD3 (E4T1B) (Cat# 4443, 1:1000) was purchased from CST.

Brefeldin A (BFA) (Cat# 420601) was purchased from eBioscience. Tamoxifen (TMX) (Cat# T5648-1G), ionomycin (Cat# I3909), and PMA (Cat# P1585-1MG) were from Sigma-Aldrich. Recombinant mouse IL-7 (Cat# 217-17) and SCF (Cat# 250-03) were from PEPROTECH. Recombinant mouse TGF-β1 (Cat# 763102) was from Biolegend. SB431542 (Cat# S1067) from Selleck was used as a TGF-βR1 inhibitor.

### Animals
WT C57BL/6J mice (body weight 15–20 g, 2–8weeks old) were purchased from Beijing Vital River Laboratory Animal Technology Co., Ltd, China. Both female and male mice were used in our experiments. Mice were maintained in specific-pathogen-free conditions with approval by the Institutional Committee of Institute of Microbiology,

Chinese Academy of Sciences. Age- and sex-matched littermates were used for all experiments. The food of all animals accorded with standard diet for rodents. Female germ-free C57BL/6J mice (3–6 weeks old) were purchased from Department of Laboratory Animal Science, Peking University Health Science Center. Gnotobiotic C57BL/6J mice were maintained in germ-free isocages and fed with sterile food and water in the germ-free animal facility of Peking University. NOD.CB17-*Prkdc*$^{scid}$*Il2rg*$^{tm1}$/Bcgen (B-NDG) mice were purchased from Biocytogen Pharmaceuticals Co., Ltd, China. *Rorc-GFP* mice (a gift from Dr. Xiao-huan Guo, Tsinghua University) were maintained in specific-pathogen-free conditions. The animals used in study were compliant with all relevant ethical regulations regarding animal research.

To generate TET1 deletion in ILCPs, *Tet1*$^{flox/+}$ mice were from Shanghai Model Organisms Center, Inc. and crossed with *Zbtb16-Cre* mice. *Tet1*$^{flox/+}$;*Zbtb16-Cre* were crossed with *Tet1*$^{flox/+}$ mice to obtain *Tet1*$^{flox/flox}$;*Zbtb16-Cre* mice. Age- and sex-matched *Zbtb16-Cre* mice served as littermate control. *Tet1*$^{flox/flox}$;*Id2-Cre* mice were generated with a similar strategy. To generate inducible TET1 deletion in CHILP, *Tet1*$^{flox/flox}$;*Id2-Cre* were treated with tamoxifen (TMX) (50 mg/kg i.p. for five consecutive days).

In vivo assay, 2 weeks old C57BL/6J mice were treated with or without 100 ng TGF-β and/or 40 ng TGF-βR1 inhibitor (SB431542) by intraperitoneal injection for 2 weeks. For cholic acid analysis, 2 weeks old C57BL/6J mice were orally gavaged with or without cholic acid (30 mg/kg) and/or TGR5 inhibitor (SBI-115, 15 mg/kg) for 2 weeks. PBS treatment served as control (Ctrl) group. For the antibiotic treatment experiment, C57BL/6 mice were oral gavaged with 100 μL of PBS (Ctrl) or antibiotic mixture (ABX, 0.05 g/mL ampicillin, vancomycin, metronidazole, neomycin and streptomycin sulfate) at PND7 for 14 days.

### Intestinal inflammation mouse model and *S.* Typhimurium infection
For DSS-induced colitis, *Zbtb16-Cre* and *Tet1*$^{flox/flox}$;*Zbtb16-Cre* mice (6–8 weeks old, body weight 15–20 g) were treated with 3% DSS in drinking water for 7 days. Mice were monitored every day.

For *S.* Typhimurium infection, *Salmonella enterica* serovar Typhimurium (*S.* Typhimurium) was from China General Microbiological Culture Collection Center. Streptomycin-resistant *S.*Typhimurium strain was selected from streptomycin plates. C57BL/6J mice (body weight 15–20 g, 6–8weeks old) were fasted for 4 h and treated with 100 μL streptomycin (200 mg/mL) 24 h before infection. Streptomycin-resistant *S.*Typhimurium strain was cultured under aerobic conditions at 37 °C for 48 h. Cell density was estimated using blood counting chamber. Bacteria were diluted into $5 \times 10^7$ CFU/mL using PBS. Each mouse was infected by oral gavage of 200 μL bacteria dilution. Feces of mice were collected at 0, 6, 12, 24, 48, and 72 h after infection and used for further analysis. *Zbtb16-Cre* and

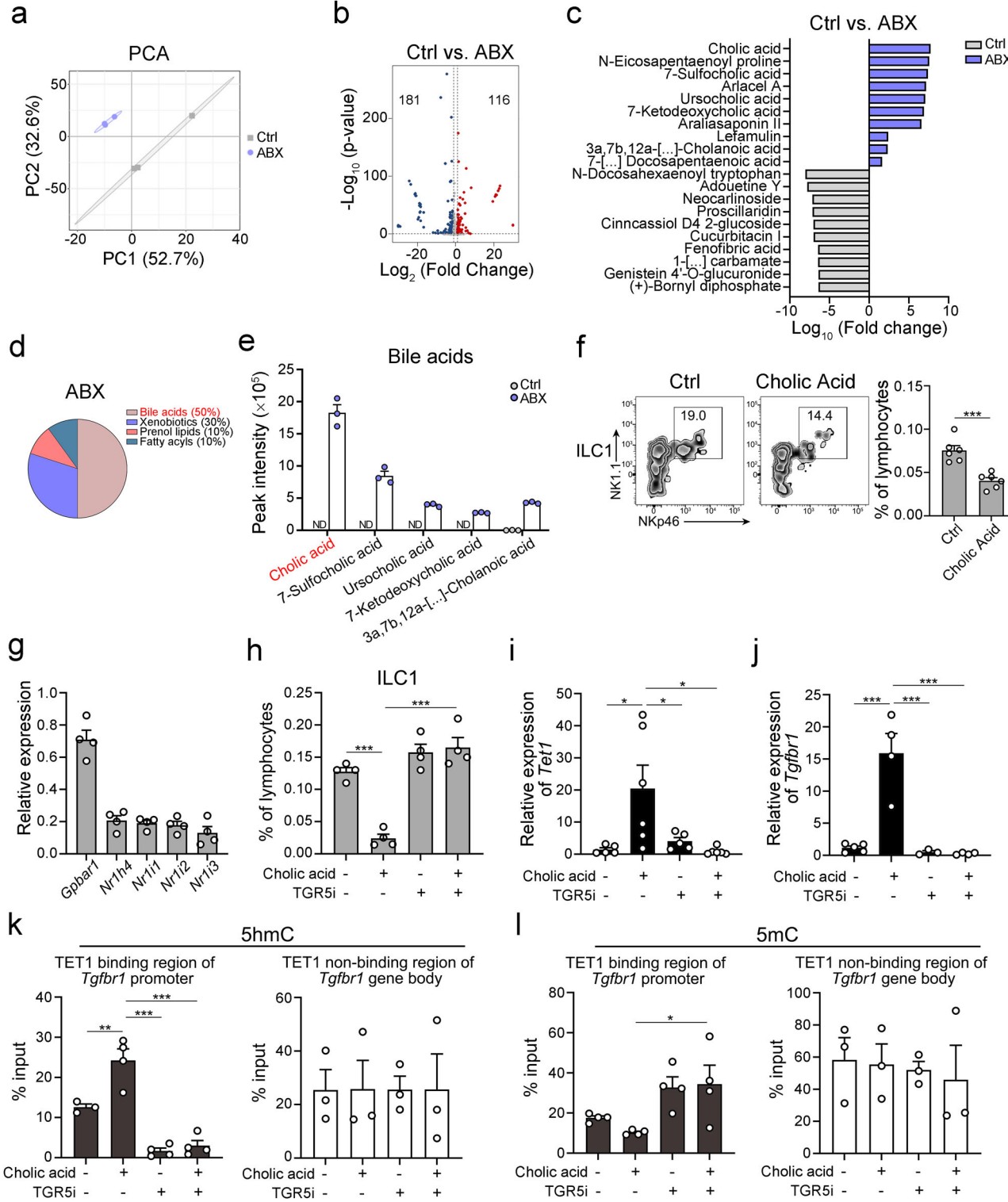

*Tet1[flox/flox];Zbtb16-Cre* mice infected with *S*.Typhimurium were performed with a similar strategy.

### Flow cytometry

Intestines tissues from mice were cut open longitudinally and Peyer's patches were removed. Next, intestines were cut into small pieces for removing epithelial layers by incubation two times in 5 mM EDTA $Ca^{2+}$ and $Mg^{2+}$ free Hank's medium for 10 min each at 37 °C, and the supernatants were collected for intraepithelial lymphocyte (IEL) analysis. Intestines tissues were then collected and cut into smaller pieces

(1–2 mm³), followed by digesting for 60 min at 37 °C with Collagenase II and III (1 mg/mL; Worthington), DNase I (200 mg/mL; Roche) on a rocking platform. The supernatants of digestive fluid were passed through a 100-μm cell strainer for removing undigested tissues pieces. The filtered fluid was collected in 50 mL tube and centrifuged at 450 g for 5 min. The centrifuged cells were washed and resuspended with 1 mL FACS buffer (0.5% FBS of 1×PBS) for the following staining of antibodies.

Cell suspensions of liver or mesenteric lymph nodes (MLNs) were obtained by passing the tissues through a 100-μm cell strainer.

**Fig. 6 | Intestinal microbiota promotes ILC1 differentiation through down-regulation of bile acids. a** Principal component analysis (PCA) for metabolome analysis of small intestinal contents from the indicated mice. Small intestinal contents were collected from mice after ABX treatment. PBS (vehicle) treatment served as a negative control (n = 3 per group). Untargeted metabolomic analysis of small intestine contents was performed by using ultrahigh-performance liquid chromatography–tandem mass spectroscopy (UPLC–MS/MS). **b** Volcano plot displayed the up- and down-regulated metabolites of ABX treatment group versus control group. The metabolites with $p < 0.05$ ($p$ values were calculated by two-sided unpaired Student's $t$ test) and $\log_2$ (fold change) >1 were considered as significantly differential metabolites. Red dot, upregulated metabolites in ABX-treated group; blue dot, upregulated metabolites in PBS-treated (Ctrl) group. **c** Comparison of metabolites abundance in the PBS (Ctrl) and ABX-treated groups. Bar plot displayed the ratios of the top 10 enriched metabolites in the Ctrl (gray) and ABX (purple) groups. **d** Pie chart showed the classes of top 10 enriched metabolites in ABX-treated group. The most enriched class of metabolites is shown in red color. **e** The levels of enriched bile acids in small intestinal contents of mice after ABX treatment. The most enriched bile acids in small intestinal contents of mice after ABX treatment were picked out and shown in bar plot. The peak intensity of the indicated bile acids was shown as the mean ± SEM. ND, not detectable. n = 3 of independent biological replicates for each group. The most abundant bile acid was highlighted in red color. **f** Cholic acid suppressed ILC1 differentiation from ILCPs in vitro. ILCPs were isolated from the bone marrow (BM) of C57BL/6 mice and cultured in medium containing 200 ng/mL IL-7 and SCF on OP9-DL1 stromal cells in the presence or absence of 100 μM cholic acid for 7 days. PBS treatment served as control (Ctrl) group. The frequency of ILC1s was analyzed by flow cytometry and shown as the mean ± SEM. ***$p < 0.001$ by unpaired Student's $t$ test ($p = 0.0004$). n = 6 for each group. **g** Analysis of bile acid receptor-related genes expression in ILCPs by qPCR. ILCPs were isolated from WT C57BL/6 mice and subjected to qPCR.

The relative expression of the indicated genes was shown as ± SEM. n = 4 for each group. **h** Cholic acid suppressed ILC1 differentiation via TGR5 signaling in vitro. ILCPs were isolated from the BM and cultured in medium containing 200 ng/mL IL-7 and SCF on OP9-DL1 stromal cells in the presence or absence of 20 ng/mL cholic acid and/or 100 μM TGR5 inhibitor (SBI-115) for 7 days. The frequency of ILC1s was analyzed by flow cytometry and shown as the mean ± SEM. ***$p < 0.001$ by one-way ANOVA. n = 4 for each group. **i, j** Cholic acid promoted *Tet1* and *Tgfbr1* expression in ILCPs. ILCPs were isolated from WT mice indicated treatment for 7 days in vitro. The relative expression of *Tet1* (**i**) (n = 5 for control group, n = 6 for cholic acid treatment group, n = 5 for TGR5 inhibitor treatment group, n = 6 for cholic acid and TGR5 inhibitor treatment group) and *Tgfbr1* (**j**) (n = 5 for control group, n = 4 for cholic acid treatment group, n = 3 for TGR5 inhibitor treatment group, n = 4 for cholic acid and TGR5 inhibitor treatment group) was analyzed by qPCR and shown as the mean ± SEM. *$p < 0.05$; **$p < 0.01$, ***$p < 0.001$ by one-way ANOVA ($p = 0.019$, 0.048, 0.011 for **i**, $p < 0.0001$ for **j**). **k** Analysis of the hydroxymethylation levels of *Tgfbr1* gene in ILCPs. ILCPs were isolated from BM of WT mice with indicated treatment for 7 days in vitro. The hydroxymethylation level in promoter (n = 3 for control group, n = 4 for cholic acid treatment group, n = 4 for TGR5 inhibitor treatment group, n = 4 for cholic acid and TGR5 inhibitor treatment group) (left panel) and gene body regions (n = 3 for each group) (right panel) of *Tgfbr1* gene were analyzed by hMeDIP-qPCR, and shown as the mean ± SEM. **$p < 0.01$; ***$p < 0.001$ by one-way ANOVA ($p = 0.0045$, <0.0001, <0.0001). **l** Analysis of the methylation levels of *Tgfbr1* gene in ILCPs. ILCPs were isolated from BM of WT mice with indicated treatment for 7 days in vitro. The methylation level in promoter (n = 4 for each group) (left panel) and gene body regions (n = 3 for each group) (right panel) of *Tgfbr1* gene were analyzed by MeDIP-qPCR, and shown as the mean ± SEM. *$p < 0.05$ by one-way ANOVA ($p = 0.036$). Source data are provided as a Source Data file.

---

Lung tissue was pre-digested with Collagenase II and III (1 mg/mL; Worthington), DNase I (200 mg/mL; Roche) and passed through a 100-μm cell strainer for obtaining a single cell suspension. Mouse bone marrow was collected from femurs by flushing with 1 mL FACS buffer, followed by passing the tissues through a 100-μm cell strainer for collecting cell suspensions. Cell suspensions were used for antibodies staining.

Cell surface markers (i.e., CD3, CD19, CD45, CD127, NK1.1, NKp46, KLRG1, ST2, CD49a, c-Kit, α4β7, PD1) were stained on ice for 60 min. For intracellular cytokine detection, cells were cultured in completed RPMI1640 media supplemented with ionomycin (500 ng/mL), PMA (50 ng/mL), and Brefeldin A (10 μg/mL) at 37 °C for 4 h. Next, cells were collected for surface marker staining, and then fixed and permeablized by Intracellular Fixation & Permeablization buffer set (eBioscience), followed by intracellular antigen (i.e., IFN-γ and RORγt) staining. Analysis of cell samples was performed on flow cytometer (FACS Aria III, BD) and Flowjo (V10) was used to analyze the flow cytometry data.

## Immunofluorescence

For immunofluorescence of ILC1s, intestines tissues were fixed in 4% paraformaldehyde (PFA) (Sigma) solution overnight and embedded in paraffin for 4 μm thick sections. Paraffin-embedded tissue slides were deparaffinized with xylenes, rehydrated through graded alcohols, and rinsed with ddH2O. Next, sections were subjected to immunohistochemistry assay by using PANO 7-plex IHC kits (PANOVUE) according to the manufacturer's description and our previous study[51]. Briefly, the sections were placed in citrate antigen retrieval buffer (pH 6.0) at 95 °C for 15 min then blocked in 10% donkey serum for 10 min. Next, primary antibodies were used for section staining for 1 h at room temperature (RT). HRP polymer antibodies and TSA dyes were incubated subsequently after brief washes. The same procedures were performed in the following staining cycles. After last round of staining, nuclei were stained by DAPI. Mounted Slides were imaged on confocal microscopy (Leica SP8).

## Real time PCR assay

RNA from different ILC subsets was extracted with Trizol reagent (Invitrogen), and complementary DNA was synthesized using the FastKing RT Kit (TIANGEN) according to manufacturer's instructions. DNA from faeces of mice was extracted with the QIAamp DNA Mini Kit (QIAGEN) according to the manufacturer's instructions. The abundance of total bacteria (16s) and expression of *Tet1*, *Tet2*, *Tet3*, *Tgfbr1*, *Tgfbr2*, *Tgfbr3*, *Gpbar1*, *Nr1h4*, *Nr1i1*, *Nr1i2* and *Nr1i3* were detected by qPCR. qPCR used SuperReal PreMix Plus Kit (SYBR Green) (TIANGEN) according to the manufacturer's instructions and was performed on the QuantStudio 7 (Thermo Fisher Scientific). Primers for real time PCR in this study are as shown in Supplementary Table 1.

## ChIP-qPCR

TET1 ChIP-qPCR was performed with ChIP-IT High Sensitivity kit (Active Motif) according to manufacturer's instructions and our previous study[52]. In brief, ILC1Ps and ILCPs were sorted by flow cytometer and fixed with formaldehyde buffer, which cross-links and preserves protein/DNA interactions. DNA was then sheared into small fragments using sonication for 10 min and incubated with 4 μg anti-TET1 antibody (Active Motif) overnight at 4 °C. The antibody-bound protein/DNA complexes were immunoprecipitated with 30 μL Protein G agarose beads for 3 h and then washed complexes via gravity filtration. Following immunoprecipitation, cross-links were reversed and the proteins were removed by Proteinase K, and the DNA is recovered and purified with DNA purification columns. ChIP enriched DNA was used as a template for subsequent qPCR with the primers in Supplementary Table 1. Ten percent of the total genomic DNA was used as the input. The fold enrichment was shown after normalizing to the input. qPCR was performed on the QuantStudio 7 (Thermo Fisher Scientific).

## MeDIP-qPCR and hMeDIP-qPCR

Genomic DNA was extracted from ILCPs with TIANamp Micro DNA Kit (TIANGEN), and sheared into small fragments using sonication. Methylated and hydroxymethylated DNA fragments were acquired

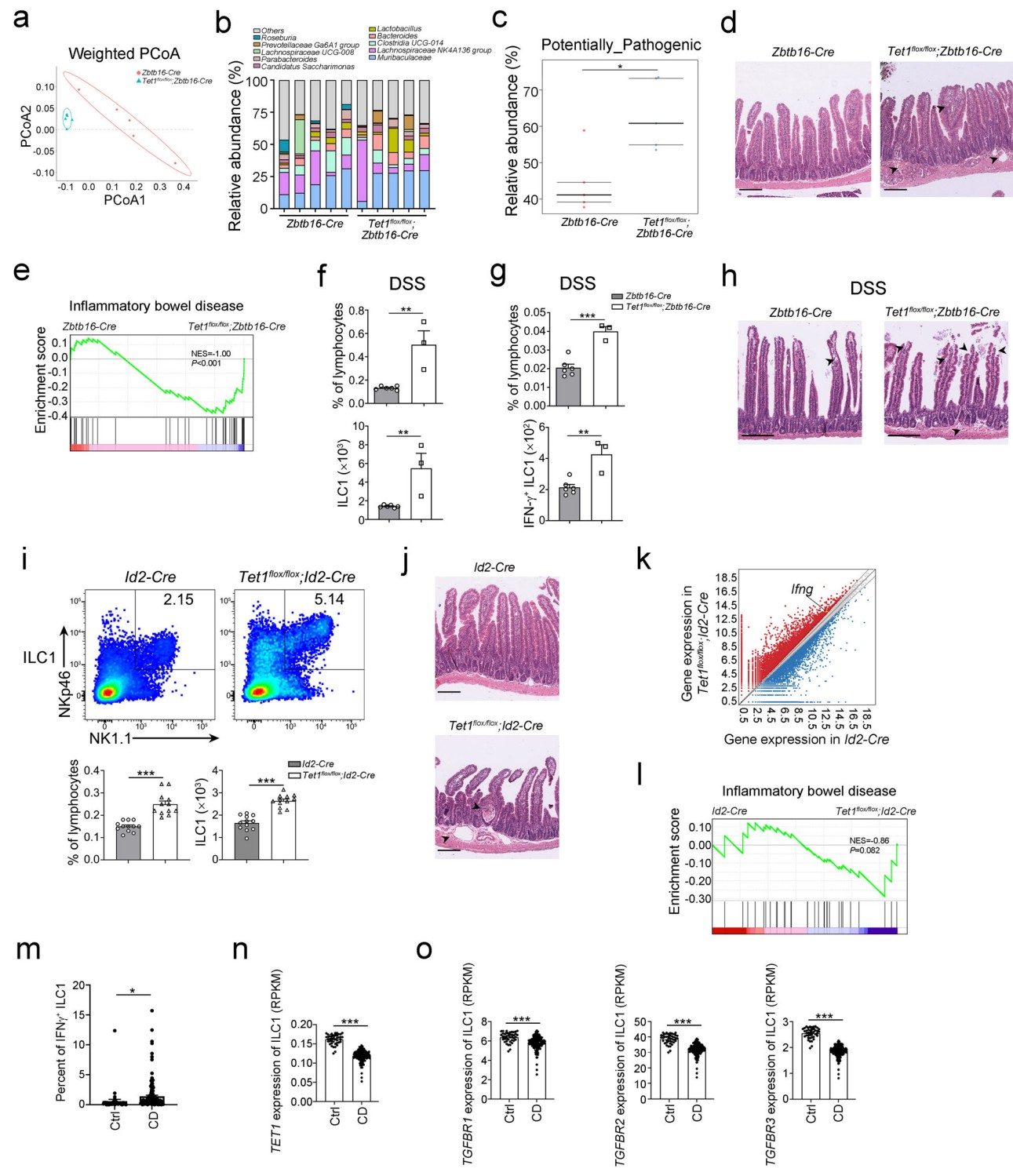

with MeDIP kit (Active Motif) and hMeDIP kit (Active Motif), respectively. The samples were incubated with 4 µL anti-5-hydroxymethylcytidine antibody at 4 °C overnight for acquiring hydroxymethylated DNA fragments. Methylated DNA fragments was pre-denatured at 95 °C for 10 min before incubation with anti-5-methylcytosine antibody. Precipitated DNA was purified with Agencourt AMPure XP Kit (Beckman). 5hmC and 5mC levels at promoters and gene body of *Tgfbr1* were further analyzed via qPCR with the designed primers in Supplementary Table 1. Ten percent of the total genomic DNA was used as the input. The fold enrichment was shown after normalizing to the input. qPCR was performed on the Quant-Studio 7 (Thermo Fisher Scientific).

**In vitro culture assay**

OP9-DL1 stroma cells (a gift from Dr. Zusen Fan (Institute of Biophysics, Chinese Academy of Sciences)) were grown in αMDM media supplemented with 10% FBS (Gibco) and 1% penicillin-streptomycin synergistic combination at 37 °C with 5% $CO_2$ for 72 h. Then, OP9-DL1 cells were digested using trypsin with EDTA from petri dishes and washed twice with sterile PBS. $5 \times 10^4$ OP9-DL1 cells were plated on

**Fig. 7 | TET1 regulates ILC1 differentiation and maintains gut homeostasis at the adult stage. a** Feces samples were collected from *Zbtb16-Cre* and *Tet1^flox/flox^;Zbtb16-Cre* mice, and subjected to 16S rRNA gene sequencing. PCoA1 and PCoA2 values were used to analyze the beta diversity of the microbiome in the gut of *Zbtb16-Cre* and *Tet1^flox/flox^;Zbtb16-Cre* mice (n = 5 for each group). **b** Bar plots show the genus levels of microbiota in feces from *Zbtb16-Cre* mice and *Tet1^flox/flox^;Zbtb16-Cre* mice. The bars represented the abundance of bacteria at the genus level in each mouse. The most abundant genera of bacteria were plotted. **c** Functional prediction of the microbiota in the intestine of *Zbtb16-Cre* and *Tet1^flox/flox^;Zbtb16-Cre* mice by using the Bugbase package of R. *$p < 0.05$ by two-sided unpaired Student's $t$ test ($p = 0.012$). n = 5 for each group. **d** H&E staining of the small intestine of *Zbtb16-Cre* and *Tet1^flox/flox^;Zbtb16-Cre* mice. The arrow heads indicated the inflammatory lesions. Scale bar, 200 μm. Data are representative of at least three independent experiments. **e** Gene set enrichment analysis (GSEA) of enriched genes in ILC1s of *Tet1^flox/flox^;Zbtb16-Cre* mice compared with *Zbtb16-Cre* mice. Gene sets of inflammatory bowel disease from the Explore the Molecular Signatures Database (MSigDB) were used. $p$ values were calculated by two-sided permutation test. **f, g** *Zbtb16-Cre* mice and *Tet1^flox/flox^;Zbtb16-Cre* mice were treated with 3% DSS in drinking water for 7 days, and the percentage and cell number of ILC1s (**f**) and IFN-γ^+^ ILC1s (**g**) were analyzed by flow cytometry. Data were shown as the mean ± SEM. **$p < 0.01$; ***$p < 0.001$ by unpaired Student's $t$ test ($p = 0.0021$, 0.0058 for **f**, $p = 0.0005$, 0.0033 for **g**). (n = 6 for *Zbtb16-Cre* mice, n = 3 for *Tet1^flox/flox^;Zbtb16-Cre* mice). **h** H&E staining of the small intestine of *Zbtb16-Cre* mice and *Tet1^flox/flox^;Zbtb16-Cre* mice after 3% DSS treatment. The arrow heads indicated the inflammatory lesions. Scale bar, 200 μm. Data are representative of least three independent

experiments. **i** Analysis of ILC1s in the small intestine from *Id2-Cre* mice and *Tet1^flox/flox^;Id2-Cre* mice after the treatment with tamoxifen (TMX) (50 mg/kg i.p. for five consecutive days). The percentage and cell number of ILC1s from *Id2-Cre* mice and *Tet1^flox/flox^;Id2-Cre* mice were analyzed by flow cytometry and are shown as the mean ± SEM. ***$p < 0.001$ by two-sided unpaired Student's $t$ test. n = 12 for each group. **j** H&E staining of the small intestine of *Id2-Cre* mice and *Tet1^flox/flox^;Id2-Cre* mice treated as described above. The arrow heads indicated the inflammatory lesions. Scale bar, 200 μm. Data are representative of at least three independent experiments. **k** Differential gene expression of ILC1s in *Id2-Cre* mice and *Tet1^flox/flox^;Id2-Cre* mice was analyzed and shown in the MA plot. Red dots represent the genes highly expressed in the *Tet1^flox/flox^;Id2-Cre* mice, and blue dots represent the genes highly expressed in the *Id2-Cre* mice. **l** Gene set enrichment analysis (GSEA) of enriched genes in ILC1s from *Tet1^flox/flox^;Id2-Cre* mice compared with *Id2-Cre* mice. Gene sets of inflammatory bowel disease from the Explore the Molecular Signatures Database (MSigDB) were used. $p$ values were calculated by two-sided permutation test. **m** Transcriptome analysis of the percentage of IFNγ^+^ ILC1s in intestinal tissue from Crohn's disease (CD) patients (n = 174) and healthy controls (Ctrl) (n = 42). The transcriptome data were from the GEO database (GSE57945). The percentage of IFNγ^+^ ILC1s was analyzed by CIBERSORTx. Data were shown as the mean ± SEM. *$p < 0.05$ by two-sided unpaired Student's $t$ test ($p = 0.0287$). **n, o** Transcriptome analysis of the *TET1* and *TGFBR* genes in ILC1s from the intestines of Crohn's disease (CD) patients (n = 174) and healthy controls (Ctrl) (n = 42). The gene expression levels of *TET1* (**n**), *TGFBR1*, *TGFBR2* and *TGFBR3* (**o**) in ILC1s were analyzed with CIBERSORTx and shown as the mean ± SEM. ***$p < 0.001$ by two-sided unpaired Student's $t$ test. Source data are provided as a Source Data file.

96-well plates in RPMI 1640 media supplemented with 10% FBS (Gibco) and 1% penicillin-streptomycin synergistic combination at 37 °C with 5% $CO_2$ for 24 h.

ILCPs were isolate from bone marrow (BM) by flow cytometer. $5 \times 10^3$ ILCPs were plated on 96-well plates containing OP9-DL1 stomal cells. Cells were cultured in RPMI 1640 medium (10% FBS and 1% penicillin-streptomycin) supplemented with 25 ng/mL IL-7 and 25 ng/mL SCF with or without 20 ng/mL TGF-β1 and 10 μM TGF-βR1 inhibitor (SB431542). After 12 days, differentiation of ILCPs was analyzed by flow cytometry. For cholic acid analysis, the media of ILCPs were supplemented with or without 20 ng/mL cholic acid and 100 μM TGR5 antagonist (SBI-115, Selleck). After 7 days, differentiation of ILCPs was analyzed by flow cytometry.

### Adoptive cell transfer experiment

ILCPs (Lin^-^c-Kit^+^CD127^+^α4β7^+^PLZF-GFP^+^, Lin = CD3, CD19, CD11b, Gr1, Ter119, CD45R) ($1 \times 10^4$) were separated from BM of *Zbtb16-Cre* and *Tet1^flox/flox^;Zbtb16-Cre* mice (6–8 weeks old, body weight 15–20 g) and transferred into B-NDG mice for differentiation of ILCPs into ILC subsets. After 4 weeks, mice were killed for flow cytometry analysis.

### MeDIP−seq and hMeDIP-seq assay

ILC1s (ILC1s = Lin^-^CD45^+^CD127^+^NK1.1^+^NKp46^+^, Lin = CD3, CD19, CD11b, Gr1, Ter119, CD45R), ILC2s (ILC2s = Lin^-^CD45^+^CD127^+^KLRG1^+^, Lin = CD3, CD19, CD11b, Gr1, Ter119, CD45R, NK1.1) and ILC3s (ILC3s = Lin^-^CD45^+^CD127^+^RORγt-GFP^+^, Lin = CD3, CD19, CD11b, Gr1, Ter119, CD45R, NK1.1) were isolated from lamina propria of small intestine, and ILC1Ps (ILC1Ps = Lin^-^CD45^+^CD127^+^NK1.1^+^NKp46^+^CD49a^+^, Lin = CD3, CD19, CD11b, Gr1, Ter119, CD45R), ILC2Ps (ILC2Ps = Lin^-^CD45^+^ST2^+^KLRG1^-^, Lin = CD3, CD19, CD11b, Gr1, Ter119, CD45R), ILC3Ps (ILC3Ps = Lin^-^CD45^+^CD127^+^α4β7^+^RORγt-GFP^+^, Lin = CD3, CD19, CD11b, Gr1, Ter119, CD45R) and ILCPs (ILCPs = Lin^-^c-Kit^+^CD127^+^α4β7^+^PLZF-GFP^+^, Lin = CD3, CD19, CD11b, Gr1, Ter119, CD45R) were sorted from bone marrow by flow cytometer. MeDIP−seq and hMeDIP-seq libraries of each ILC subsets were generated with MeDIP kit (Active Motif) and hMeDIP kit (Active Motif) according to manufacturer's instructions. In brief, genomic DNA were extracted from cells with TIANamp Micro DNA Kit (TIANGEN), and sheared into small fragments using sonication. The samples were incubated with

4 μL anti-5-hydroxymethylcytidine antibody at 4 °C overnight for acquiring hydroxymethylated DNA fragments. Methylated DNA fragments was pre-denatured at 95 °C for 10 min before incubation with anti-5-methylcytosine antibody. Precipitated DNA was linked with P5 and P7 adapters using Library Preparation NovoNGS® Multiplex Oligos set 1 for Illumina kit (Novoprotein), and hMeDIP-seq and MeDIP−seq libraries prepared using DNA Library Prep Kit for Illumina (Novoprotein). Qualified libraries were sequenced on Illumina NovaSeq platform for generating pair-end reads. Clean reads were aligned to *M. musculus* reference genome (mm10) by using Bowtie2. Peaks were called by MACS2 and visualized by IGV, ChIPseeker, and ClusterProfiler. Annotation of peaks was performed with Homer.

### 16S rRNA sequencing and analysis

DNA was extracted from faeces of *Zbtb16-Cre* and *Tet1^flox/flox^;Zbtb16-Cre* mice with QIAamp DNA Mini Kit (QIAGEN) following manufacturer's instructions. Sample collection and preservation buffer served as negative control. Ribosomal 16S rDNA V4 region was amplified with universal primers: 515 F (5′-GTGCCAGCMGCCGCGGTAA-3′) and 806 R (5′-GGACTACHVGGGTWTCTAAT-3′). The protocol of PCR thermocycler was as follows: 1 min at 98 °C followed by 30 cycles of 10 s at 98 °C, 30 s at 50°C and 30 s at 72 °C, and a final 5 min at 72 °C. PCR reactions were performed in three replicates and purified with Gene-JET TM Gel Extraction Kit (Thermo Scientific). Sequencing libraries were generated with Ion Plus Fragment Library Kit (Thermo Scientific) The libraries were sequenced on Illumina HiSeq X Ten platform for generating 400-bp single-end reads[41].

Cutadapt (Version 1.9.1) and UCHIME algorithm were used to obtain clean reads. The clean reads were assigned to the same OTUs with ≥ 97% by using Uparse software, (version 7.0.1001) similarity. The α-diversity indexes (including ACE, Chao1, Shannon and Observed species) were calculated using MOTHUR program. QIIME 2 was used to calculate weighted-UniFrad distances of PCoA and Shannon index. The function profiles of microbiota were predicted by using Bugbase.

### RNA-sequencing and analysis

The ILC1s in lamina propria of small intestine were isolated by flow cytometer. $2 \times 10^4$ cells were harvested and extracted total RNA with TRIzol reagent (Thermo Fisher Scientific) following the standard

protocol. RNA quality and quantity was determined by NanoDrop ND-2000 (NanoDrop Technologies) and 2100 Bioanalyser (Agilent). Only high-quality RNA samples (OD260/280 = 1.8–2.2, OD260/230 ≥ 2.0, RIN ≥ 7) were selected to construct sequencing library. Then amplification of RNA was carried out using the Smart-Seq2 method. After cDNA synthesis, PCR amplification, purification, and fragmentation, the samples were used to construct Illumina library. Qualified libraries were sequenced on Illumina Hiseq X Ten platform (PE150) for generating pair-end reads.

For all statistical analysis, Fastqc was used to evaluate raw reads for quality control, followed by trimming sequence with Trimmomatic 0.39. Trimmed read were mapped to the *M. musculus* reference genome (GRCm39) with Hisat2 2.1.0. StringTie 2.2.1 was used to assemble and quantitate transcripts. Differentially expressed genes were analyzed and plotted using EdgeR 3.14 and ggplot2, respectively. Gene Set Enrichment Analysis was performed with GSEA 4.2.2.

### Metabolomics and analysis

Small intestinal contents from mice were subjected for metabolomics analysis by ultra-high performance liquid chromatography-mass spectrometry (UPLC-MS). Small intestinal contents were collected from mice after PBS and ABX treatment. Acquired samples were snap-frozen in liquid nitrogen and stored at −80 °C for following analysis. For UPLC-MS analysis, 50 mg of small intestinal contents were mixed with 400 μL extracting solution (acetonitrile:methanol: 1:1, v/v), vigorously shaken for 30 min, and incubated on ice for 5 min. After centrifugation (12,000 $g$ at 4 °C for 15 min), the supernatant was loaded for the analysis performing ExionLCTMAD system (AB Sciex, USA) and ACQUITY UPLC BEH Amide reverse phase column (2.1 mm id × 100 mm × 1.7 mm) (Waters, USA). After loading, the samples were filtered and centrifuged to remove the particle. The mixture of 25 mM of acetic acid, ammonia water, and acetonitrile were used in the mobile phases: A and B, respectively. The elution was eluted with 5% A–95% B for 0.5 min, 35% A–65% B for 6.5 min, 60% A–40% B for 2 min, and then eluted with 5% A–95% B for 3.5 min to balance the column. The total chromatographic elution process was 12 min, with the flow rate of 500 μL/min. AB 5600 Triple TOF mass spectrometer system (AB Sciex, USA) with Analyst TF 1.7 software was used to screen molecule and ion >100 and collect secondary mass spectrometric data. The Progenesis QI 2.3 (Nonlinear Dynamics, Waters, USA) software was used for raw peak exacting, data baseline filtering and calibration, peak alignment, peak identification, and peak area integration. PCA showed the distribution of origin data. The significantly different metabolites between comparable groups were identified with log$_2$ (fold change) value more than 1 and $p$ value less than 0.05. HMDB and KEGG database was utilized to annotate the classes of metabolites.

### Statistics and reproducibility

Animal experiments were repeated at least two times with similar results, no statistical method was used to predetermine sample size. For statistical analysis, data were analyzed by using GraphPad Prism 9.0. Two-sided unpaired Student's parametric $t$ test or One way ANOVA were used according to the type of experiments. $p ≤ 0.05$ were considered significant (*$p < 0.05$; **$p < 0.01$; ***$p < 0.001$); $p > 0.05$, non-significant (ns). All flow cytometry data were analyzed with FlowJo (Treestar).

### Reporting summary

Further information on research design is available in the Nature Portfolio Reporting Summary linked to this article.

## Data availability

The sequence data generated in this study have been deposited to China National Microbiology Data Center (NMDC) (https://nmdc.cn/resource/) under accession numbers: PRJCA020429 and PRJCA025083. All other data are available within the Article and Supplementary Files. Source data are provided with this paper.

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

## Acknowledgements

We thank Drs. Chun-Jun Guo (Cornell University, USA), Zhuqiang Zhang (Institute of Biophysics, Chinese Academy of Sciences, China) and Pengyan Xia (Peking University, China) for technical suggestion and support. We thank Tong Zhao (Institute of Microbiology, Chinese Academy of Sciences) and Kuo Zhang (Department of Laboratory Animal Science, Health Science Center, Peking University) for technical support. This work was supported by Key Research Program of Frontier Sciences of Chinese Academy of Sciences (ZDBS-LY-SM025), the National Key R&D Program of China (2021YFA1300202, 2023YFC2306200, 2022YFC2302900), National Natural Science Foundation of China (92169113), and CAS Project for Young Scientists in Basic Research (YSBR-010). Cartoons in Fig. 2d is created with BioRender.com and released under a Creative Commons Attribution-NonCommercial-NoDerivs 4.0 International license.

## Author contributions

X.Z. performed experiments and analyzed data; X.G., Z.L., F.S., D.Y., M.Z., and X.Q. performed experiments. S.W. initiated the study, and organized, designed, and wrote the paper.

## Competing interests

The authors declare no competing interests.
