## [Peer Review File · Nature Communications]

Microbiota regulates the TET1-mediated DNA hydroxymethylation program in murine innate lymphoid cell differentiationEditorial Note: Due to reviewer changes during the peer review of this manuscript and for consistency between the rounds of review, there is no reviewer 2 in this instance, and the remaining reviewers remain numbered as reviewers 1 and 3.

REVIEWER COMMENTS

Reviewer #1 (Remarks to the Author):

This is a very nice research paper that shows the role of the microbiome and especially the bile salt cholic acid in TET1 expression and ILC1 differentiation in the gut. Cholic acid induces TET1 expression, which inhibits the differentiation into ILC1s from ILCPs. Furthermore an interesting observation into potential clinical application was made, as TET1-deficient mice show higher amounts of intestinal ILC1s and more intestinal inflammation, which correlated with gene sets seen in inflammatory bowel disease. Comparable observations (less TET1 and more ILC1s) were done in the intestine of Crohn's disease patients.

Overall this paper is well written, has clear and informative figures and a clear message. To my opinion the paper could be published in its current form. I only have some minor issues that could be addressed:

*To improve understanding of the figures for a reader who is not confronted with ILC markers on a daily basis (such as me), some extra marks/indicators could be of help. I give a few examples:

-Figure 2 is an example where the extra indicators are very helpful (e.g. ILC1-3 in panel a) -> could this also be applied to the other figures? E.g.:

-The labeling of the last panel in fig 2d and 2e by adding that this concerns ILC1s

-Fig 4a,h

-Ext data fig 5

-Etc.

*Line 189, could it either in the text or in the figure the weaning time point be included.

*In Fig 6c apart from Cholic acid, other metabolites are differentially regulated as well. Could these also be involved in ILC regulation? Is there anything known already in literature? I was intrigued by tryptophan in this panel, as this is also a central regulator in lymphocyte differentiation. It would be good to add a few extra lines on this to the discussion.

Reviewer #3 (Remarks to the Author):

Zhang et al.'s manuscript examines DNA methylation and hydroxymethylation in innate lymphoid cells (ILCs) and describes the role of TET1 in ILC1 development. They utilized immunoprecipitation-based methods to profile the DNA methylation and hydroxymethylation landscape of ILC and ILC precursor subsets. They focused specifically on TET1, one of the TET-family enzymes which catalyze conversion of methylated to hydroxymethylated DNA. Using mice where TET1 is ablated in ILC precursor, they demonstrate a bile acid-microbiota-TET1-TGFBR1 axis in restricting gut ILC1 development. They also demonstrate that although loss of TET1 early on results in increased ILC1 development, loss of TET1 in adulthood exacerbates colitis via impairment of ILC1 function.

This manuscript is interesting since it presents a novel mechanism that relates to known

concepts about microbiota-ILC crosstalk and epigenetic regulation of ILC development (1). It also confirms and expands on recent work focused on DNA hydroxymethylation in ILCs (2). Additionally, this study was able to zoom in on a particular pathway known to play important roles in ILC development and function and connect it to novel epigenomic findings (3,4). While the mechanisms proposed are potentially important in further understanding the multidisciplinary nature of ILC1 development, the study is lacking in its characterization of DNA methylation and hydroxymethylation during ILC differentiation. It should also investigate and discuss the roles of TET1 with regards to other epigenomic factors and gene expression. Moreover, the study relies on one line of evidence in multiple occasions (in vitro experiments with TGFB and TGFB_{Ri}, as well as with cholic acid, for example). Taken together, the authors present an interesting epigenetic mechanism that regulates ILC1 development. The study and its claims will be much better substantiated following some revisions.

Major issues:

1. The authors discussed mainly promoters in their analysis of DNA methylation/hydroxymethylation. However, DNA methylation/hydroxymethylation occurs throughout the genome, and their presence in non-promoter CpGs also contribute to epigenetic regulation. Can the authors further discuss DNA methylation and hydroxymethylation of gene bodies, as well as distal cis-regulatory elements (CREs), particularly with regards to ILCP-ILC1/2/3 transition? What transcription factor binding sites are being preferentially hydroxymethylated in this transition? DNA hydroxymethylation has been proposed to regulate gene expression independent of its role in demethylation. In ILCP-ILC transitions, which genes' promoters are completely demethylated (i.e. no longer detected in both MeDIP and hMeDIP), and which are hydroxymethylated? How does the expression of these genes differ? Is there a difference in the transcription factor binding motifs enriched in completely demethylated vs hydroxymethylated CREs?
2. TET1 has been shown to play a role in preventing methylation spreading into CGIs (5). How does ablation of TET1 in ILCP affect CGI methylation in ILC subsets? Additionally, how does loss of TET1 affect the chromatin landscape and enhancer usage in these populations? Are there changes in gene expression, and if so, what relationship is there between gene expression and epigenetic changes in this case?
3. The authors compared the 5hmC and 5mC landscape of ILC1 and ILCPs in germ free, SPF, and antibiotics treated conditions. How do the changes in 5mC and 5hmC relate with other epigenetic changes (such as chromatin accessibility and enhancer usage) and gene expression under these circumstances?
4. The claims regarding TGF-beta signaling and bile acid signaling are only substantiated by in vitro experiments using pharmacologic methods. Can the authors demonstrate some of these results in vivo? Additionally, loss of function in vitro experiments would greatly support these conclusions.

Minor issues:

1. The authors show that antibiotic treatment results in decreased gut ILC1 development. Does antibiotic treatment alter ILC1 development in other organs? The discussion mentioned that it doesn't, but no data is shown.
2. The authors present very interesting data on the alteration of the microbiome in absence of TET1 in ILCPs. Can the authors comment on the potential mechanisms that may contribute to the expansion of specific bacterial strains? Is this phenomenon due to ILC1-intrinsic changes, or could ILC2 and ILC3 also play a role? It would be important to compare the gene expression of these subsets (as mentioned in major point #2).
3. CUT&Tag is recommended for profiling histone modifications, but not necessarily other

DNA-binding proteins as the targets may not be abundant enough for use with such low input (10000 cells as used in this study). The tracks in figures 1h and 3h showed scattered reads and not necessarily peaks, which appears consistent with a failed CUT&Tag experiment. Unless the authors can perform quality control for this experiment by titrating cell count using this antibody, this reviewer thinks they can simply omit this data.

4. In lines 300-302 in the discussion, the authors stated "The 5mc/5hmc modification of the *Tgfbr1* gene by TET1 suppresses TGF- β signaling and plays an important role in the differentiation of ILC1s." The authors might want to change this sentence since the manuscript suggests that TET1 augments TGFB signaling, which suppresses ILC1 development.

References

- 1 Gury-BenAri, M. et al. The spectrum and regulatory landscape of intestinal innate lymphoid cells are shaped by the microbiome. *Cell* 166, 1231-1246. e1213 (2016).
- 2 Peng, V. et al. Whole-genome profiling of DNA methylation and hydroxymethylation identifies distinct regulatory programs among innate lymphocytes. *Nature immunology* 23, 619-631 (2022).
- 3 Wang, L. et al. TGF- β induces ST2 and programs ILC2 development. *Nature Communications* 11, 35 (2020).
- 4 Cortez, V. S. et al. Transforming growth factor- β signaling guides the differentiation of innate lymphoid cells in salivary glands. *Immunity* 44, 1127-1139 (2016).
- 5 Jin, C. et al. TET1 is a maintenance DNA demethylase that prevents methylation spreading in differentiated cells. *Nucleic acids research* 42, 6956-6971 (2014).

Point-by-point response to the reviewers

Reviewer #1

This is a very nice research paper that shows the role of the microbiome and especially the bile salt cholic acid in TET1 expression and ILC1 differentiation in the gut. Cholic acid induces TET1 expression, which inhibits the differentiation into ILC1s from ILCPs. Furthermore an interesting observation into potential clinical application was made, as TET1-deficient mice show higher amounts of intestinal ILC1s and more intestinal inflammation, which correlated with gene sets seen in inflammatory bowel disease. Comparable observations (less TET1 and more ILC1s) were done in the intestine of Crohn's disease patients. Overall this paper is well written, has clear and informative figures and a clear message. To my opinion the paper could be published in its current form. I only have some minor issues that could be addressed:

Minor issues

1. To improve understanding of the figures for a reader who is not confronted with ILC markers on a daily basis (such as me), some extra marks/indicators could be of help. I give a few examples: -Figure 2 is an example where the extra indicators are very helpful (e.g. ILC1-3 in panel a) -> could this also be applied to the other figures? E.g.: -The labeling of the last panel in fig 2d and 2e by adding that this concerns ILC1s; -Fig 4a,h; -Ext data fig 5; -Etc.

Answer: We appreciate this comment from this reviewer. We have marked the ILCs in our new version of figures.

2. Line 189, could it either in the text or in the figure the weaning time point be included.

Answer: Thanks very much for the suggestion. We have added the weaning time point at Fig 4a and line 189 in our new version.

3. In Fig 6c apart from Cholic acid, other metabolites are differentially regulated as well. Could these also be involved in ILC regulation? Is there anything known already in literature? I was intrigued by tryptophan in this panel, as this is also a central regulator in lymphocyte differentiation. It would be good to add a few extra lines on this to the discussion.

Answer: This is a very good point. Thanks very much for the suggestion. In this study, we have analyzed composition changes of metabolites in small intestinal contents after

antibiotic treatment. The levels of bile acids were increased after antibiotic treatment, especially the primary bile acid. Other metabolites (such as N-Eicosapentaenoyl proline, Arlacel A, Araliasaponin II, Lefamulin and 7,10,13,16,19-Docosapentaenoic acid) also increased in antibiotic-treated mice, but no studies have reported the involvement of these metabolites in ILC regulation. Just as this reviewer mentioned that tryptophan enables to modulate lymphocyte activation via the production of indole-3-aldehyde (I3A) ^{1,2}. I3A can activate ILC3s via AhR-dependent pathway ³. In our study, tryptophan and I3A in small intestinal content could not be detected at weaning stage. The role of tryptophan in regulating ILC1 differentiation is worthy to be studied in the future work. We have added these in “Discussion” section and highlighted in yellow.

Reviewer #3:

Zhang et al.'s manuscript examines DNA methylation and hydroxymethylation in innate lymphoid cells (ILCs) and describes the role of TET1 in ILC1 development. They utilized immunoprecipitation-based methods to profile the DNA methylation and hydroxymethylation landscape of ILC and ILC precursor subsets. They focused specifically on TET1, one of the TET-family enzymes which catalyze conversion of methylated to hydroxymethylated DNA. Using mice where TET1 is ablated in ILC precursor, they demonstrate a bile acid-microbiota-TET1-TGFBR1 axis in restricting gut ILC1 development. They also demonstrate that although loss of TET1 early on results in increased ILC1 development, loss of TET1 in adulthood exacerbates colitis via impairment of ILC1 function. This manuscript is interesting since it presents a novel mechanism that relates to known concepts about microbiota-ILC crosstalk and epigenetic regulation of ILC development (1). It also confirms and expands on recent work focused on DNA hydroxymethylation in ILCs (2). Additionally, this study was able to zone in on a particular pathway known to play important roles in ILC development and function and connect it to novel epigenomic findings (3,4). While the mechanisms proposed are potentially important in further understanding the multidisciplinary nature of ILC1 development, the study is lacking in its characterization of DNA methylation and hydroxymethylation during ILC differentiation. It should also investigate and discuss the roles of TET1 with regards to other epigenomic factors and gene expression. Moreover, the study relies on one line of evidence in multiple occasions (in vitro experiments with TGF β and TGF β Ri, as well as with cholic acid, for example). Taken together, the authors present an interesting epigenetic mechanism that regulates ILC1 development. The study and its claims will be much better substantiated following some revisions.

Answer: We appreciated the comments from this reviewer. We have tried our best to improve our manuscript according to the comments by reviewer.

Major issues:

1. The authors discussed mainly promoters in their analysis of DNA methylation/hydroxymethylation. However, DNA methylation/hydroxymethylation occurs

throughout the genome, and their presence in non-promoter CpGs also contribute to epigenetic regulation. Can the authors further discuss DNA methylation and hydroxymethylation of gene bodies, as well as distal cis-regulatory elements (CREs), particularly with regards to ILCP-ILC1/2/3 transition? What transcription factor binding sites are being preferentially hydroxymethylated in this transition? DNA hydroxymethylation has been proposed to regulate gene expression independent of its role in demethylation. In ILCP-ILC transitions, which genes' promoters are completely demethylated (i.e. no longer detected in both MeDIP and hMeDIP), and which are hydroxymethylated? How does the expression of these genes differ? Is there a difference in the transcription factor binding motifs enriched in completely demethylated vs hydroxymethylated CREs?

Answer: We appreciated the comments from this reviewer. These are very good points. We've analyzed them and answered them one by one below:

1) We have analyzed DNA methylation and hydroxymethylation level of gene bodies and distal cis-regulatory elements (CREs) (enhancers) during ILCP-ILC1/2/3 transition (Attached Fig. 1a-d). Our previous data showed that promoter undergo obvious changes during ILCP-ILC1/2/3 transition (Fig. 1e). However, the methylation and hydroxymethylation levels of most gene bodies and enhancers remained unchanged during the differentiation of ILCPs into ILC2s and ILC3s (Attached Fig. 1a-d), suggesting that the elements such as gene bodies and CREs have less changes in methylation and hydroxymethylation compared to the promoters.

2) We next analyzed the 5hmC distributions in lineage-specific transcription factors (i.e. PLZF, ID2, TOX, T-bet, GATA3, ROR α and ROR γ t) binding sites of ILC subsets (Attached file 1). GATA3 binding motifs in ILC2s undergo upregulation of hydroxymethylation, and PLZF and ID2 binding motifs in ILCs undergo downregulation of hydroxymethylation during ILCP-ILC1/2/3 transition (Attached Fig. 2a-c). DNA hydroxymethylation was proposed to regulate gene expression independently of demethylation. We further analyzed the genes with hyper-hydroxymethylated promoters (Hyper-HMPs) or completely demethylated promoters (CDMPs) and during ILCP-ILC1/2/3 transition (Attached file 2 and Attached file 3), and compared them with differentially expressed genes (DEGs). During ILCP-ILC1/2/3 transition, Hyper-HMPs mainly contributed to the up-regulation of gene expression

(Attached Fig. 2d-f). We also analyzed the genes with completed demethylated promoters (CDMPs) and DEGs. The completed demethylation may not contribute to the differential expression of genes (Attached Fig. 2g). These results indicated that DNA hydroxymethylation of promoters may contribute to the DEGs during ILCP-ILC1/2/3 transition.

3) We also analyzed the enrichment of binding motifs of key transcription factors (i.e. T-bet, GATA3 and ROR γ t) in completely demethylated and hydroxymethylated promoters and enhancers (Attached Fig. 3a, b). The transcription factor binding motifs enriched in completely demethylated CREs show differences with hydroxymethylated CREs.

All of these data suggested that hyper-hydroxymethylation of promoters was crucial for the regulation of genes expression during ILC differentiation, which was consistent with our previous studies (Fig 1d-g). Therefore, we have changed the title and highlighted the hydroxymethylation program during ILC differentiation.

2. TET1 has been shown to play a role in preventing methylation spreading into CGIs (5). How does ablation of TET1 in ILCP affect CGI methylation in ILC subsets? Additionally, how does loss of TET1 affect the chromatin landscape and enhancer usage in these populations? Are there changes in gene expression, and if so, what relationship is there between gene expression and epigenetic changes in this case?

Answer: These are very good points. We have analyzed the methylation of CpG islands (CGIs), enhancers and gene bodies in ILC subsets after ablation of TET1 in ILCP (Attached Fig. 3c-h). Intriguingly, ablation of TET1 primarily affected methylation of CGIs in ILC1s (Attached Fig. 3c), while exhibiting few influences on enhancers and gene bodies (Attached Fig. 3d-e). After analysis of gene expression pattern, we found that transcriptome changes were correlated with CGIs and promoter methylation in ILC1s along with TET1 abrogation (Attached Fig. 3c-e lower panel and Fig. 3b, c). In ILC2s and ILC3s, methylation of most of CGIs, enhancers and gene bodies remained unchanged after ablation of TET1 (Attached Fig. 3f-h). Analysis of chromatin landscape (including promoter, enhancer, CGI and gene body regions) in ILC subsets showed that the loss of TET1 affected the methylation of CGIs and promoters, but showed few influences on enhancers and gene

bodies (Attached Fig. 3c-h and Fig. 3b). We have tried to analyze the enhancer usage by CAGE-seq⁴, but we failed due to the insufficient number of cells. The enhancer usage regulated by TET1 in ILCs is worthy to be studied in the future work. We also analyzed the relationship between the differential methylation gene elements and DEGs after TET1 depletion in ILCPs. Our results revealed that DEGs were mainly associated with differential methylation CGIs and promoters in ILC1s (Attached Fig. 3c-e (lower panel) and Fig. 3b, c), suggesting that TET1 affected gene expression mainly through methylation of CGIs and promoters in ILC1s.

3. The authors compared the 5hmc and 5mc landscape of ILC1 and ILCPs in germ free, SPF, and antibiotics treated conditions. How do the changes in 5mc and 5hmc relate with other epigenetic changes (such as chromatin accessibility and enhancer usage) and gene expression under these circumstances?

Answer: We appreciated the comments from this reviewer. To understand the epigenetic changes and gene expression after ablation of the microbiota, we analyzed transcriptome, DNA methylation and hydroxymethylation data of ILC1s under SPF or germ-free condition (Attached Fig. 4a-d). Notably, transcriptome changes were correlated with hyper-hydroxymethylation of promoter and enhancer regions, while methylation/hydroxymethylation changes in gene bodies regions did not show obvious influence on gene expression under germ-free conditions. As previously reported, the change of gene expression was tightly associated with the regulation of chromatin accessibility and enhancer usage^{5, 6, 7, 8}. After analysis of gene expression pattern, our results revealed that the changes of hydroxymethylation of promoter and enhancer regions were associated with DEGs (Attached Fig. 4b-d). It is possible that the change of 5hmC is related to the change of chromatin accessibility and enhancer usage. The other epigenetic changes under different circumstances are worthy to be studied in the future work.

4. The claims regarding TGF-beta signaling and bile acid signaling are only substantiated by in vitro experiments using pharmacologic methods. Can the authors demonstrate some

these results *in vivo*? Additionally, loss of function *in vitro* experiments would greatly support these conclusions.

Answer: Thanks very much for the suggestion. We have investigated TGF-beta signaling and bile acid signaling *in vivo* assays. The treatment of TGF- β 1 in mice inhibited the differentiation of ILC1s in the small intestine. The addition of TGF- β R1 inhibitor enabled to reverse the inhibition of ILC1 differentiation (Attached Fig. 4e). Similarly, cholic acid enable to inhibit ILC1 differentiation, and ILC1 population were restored by treatment with cholic acid receptor inhibitor *in vivo* (Attached Fig. 4f). These results suggested that TGF- β signaling and bile acid signaling played an important role in the differentiation of ILC1s.

Minor issues:

1. The authors show that antibiotic treatment results in decreased gut ILC1 development. Does antibiotic treatment alter ILC1 development in other organs? The discussion mentioned that it doesn't, but no data is shown.

Answer: We appreciate this reviewer for pointing out this point. We analyzed ILC1s in other organs (i.e. liver and lung) after antibiotic treatment. Our results revealed that gut microbiota contributes to the differentiation of ILC1s in the intestine but not in other tissues (new Extended Data Fig. 5c-e).

2. The authors present very interesting data on the alteration of the microbiome in absence of TET1 in ILCPs. Can the authors comment on the potential mechanisms that may contribute to the expansion of specific bacterial strains? Is this phenomenon due to ILC1-intrinsic changes, or could ILC2 and ILC3 also play a role? It would be important to compare the gene expression of these subsets (as mentioned in major point #2)

Answer: We appreciate these comments from this reviewer. These are very good points. After ablation of *Tet1* gene, the cell number of ILC2s and ILC3s were not apparently changed (Fig. 2a). We further analyzed the transcriptional changes of ILC2s and ILC3s in *Tet1^{flox/flox};Zbtb16-Cre* mice. We found that the effector genes in ILC2s and ILC3s were not changed (Attached Fig. 5a), indicating that they may not affect gut environment by their effector function. Gene set enrichment analysis (GSEA) of DEGs showed that the Th1/Th2 and Th17 differentiation pathways in ILC2s and ILC3s remained unchanged after *Tet1*

ablation (Attached Fig. 5b, c). After *Tet1* deletion, ILC1 cells showed an inflammatory state, producing more inflammatory factors. The inflammatory microenvironment may be the main reason for the impaired barrier function of the intestine, and resulted in the increase of some pathogenic bacteria. Therefore, we believe that it is mainly the intrinsic changes of ILC1 that cause the changes in the intestinal bacteria.

3. CUT&Tag is recommended for profiling histone modifications, but not necessarily other DNA-binding proteins as the targets may not be abundant enough for use with such low input (10000 cells as used in this study). The tracks in figures 1h and 3h showed scattered reads and not necessarily peaks, which appears consistent with a failed CUT&Tag experiment. Unless the authors can perform quality control for this experiment by titrating cell count using this antibody, this reviewer thinks they can simply omit this data.

Answer: Thanks very much for the suggestion. We have removed these data in our new version according to the suggestion.

4. In lines 300-302 in the discussion, the authors stated "The 5mc/5hmc modification of the *Tgfb1* gene by TET1 suppresses TGF-B signaling and plays an important role in the differentiation of ILC1s." The authors might want to change this sentence since the manuscript suggests that TET1 augments TGFB signaling, which suppresses ILC1 development.

Answer: Thanks very much for the suggestion. We have modified our description as suggested by this reviewer in our new version and highlighted in yellow.

Attached Figure 1-5

Attached Fig. 1. DNA methylation and hydroxymethylation level of gene bodies and distal cis-regulatory elements. (a) Analysis of differentially methylated gene bodies of ILCP-ILC1/2/3 transition. Differentially methylated gene bodies of ILCPs vs ILC1s, ILCPs vs ILC2s and ILCPs vs ILC3s were identified and shown in the MA plot. Red dots represent the highly methylated gene bodies in the ILCP, and blue dots represent the gene bodies highly methylated in the ILC1s, ILC2s or ILC3s. (b) Analysis of differentially hydroxymethylated gene bodies during ILCP-ILC1/2/3 transition. Differentially

hydroxymethylated gene bodies of ILCPs vs ILC1s, ILCPs vs ILC2s and ILCPs vs ILC3s were identified and shown in the MA plot. Red dots represent the highly hydroxymethylated gene bodies in the ILCP, and blue dots represent the gene bodies highly hydroxymethylated in the ILC1s, ILC2s or ILC3s. (c) Analysis of differentially methylated enhancers during ILCP-ILC1/2/3 transition. Differentially methylated enhancers of ILCPs vs ILC1s, ILCPs vs ILC2s and ILCPs vs ILC3s were identified and shown in the MA plot. Red dots represent the highly methylated enhancers in the ILCP, and blue dots represent the enhancers highly methylated in the ILC1s, ILC2s or ILC3s. (d) Analysis of differentially hydroxymethylated enhancers during ILCP-ILC1/2/3 transition. Differentially hydroxymethylated enhancers of ILCPs vs ILC1s, ILCPs vs ILC2s and ILCPs vs ILC3s were identified and shown in the MA plot. Red dots represent the highly hydroxymethylated enhancers in the ILCPs, and blue dots represent the enhancers highly hydroxymethylated in the ILC1s, ILC2s or ILC3s.

Attached Fig. 2. Hydroxymethylation and methylation regulation during ILCP-ILC transition. (a-c) Analysis of hydroxymethylation level of transcription factors binding sites in ILC subsets. The GATA3 (motif1: AGATAAGA; motif2: GATAGATA; motif3: TAGATAAA) (a), ID2 (motif4: GCATGCGC) (b) and PLZF (motif5: ACAGGAAG; motif6: CTGATTGG; motif7: GCCAATGG) (c) binding motifs were identified from ChIPBase v3.0 database. The methylation and hydroxymethylation levels of these motifs in ILC1s, ILC2s, ILC3s and ILCPs were calculated with BEDTools and HOMER software. ILC1s, ILC2s and ILC3s from lamina propria of small intestine and ILCPs from bone marrow (BM) were isolated from wild type (WT) mice followed by hMeDIP-seq. Each ILC subset was collected from at least three mice. (d-f) Analysis of hyper-hydroxymethylated promoters (Hyper-HMPs) and differentially expressed genes (DEGs) between ILC subsets and ILCPs. Venn diagrams show overlapping Hyper-HMPs and DEGs in ILC1s (d), ILC2s (e), or ILC3s (f) compared with ILCPs (upper panel). The percentages of up-regulated and down-regulated genes in

overlapping region in (upper panel) were shown (lower panel). The transcriptome data of ILCPs, ILC1s, ILC2s and ILC3s were from the GEO database (GSE161441). (g) Analysis of completely demethylated promoters (CDMPs) and differentially expressed genes (DEGs) between ILC subsets and ILCPs. Venn diagrams show overlapping CDMPs and DEGs in ILC1s (d), ILC2s (e), or ILC3s (f) compared with ILCPs. The transcriptome data of ILCPs, ILC1s, ILC2s and ILC3s were from the GEO database (GSE161441).

Attached Fig. 3. Regulation of gene methylation by TET1 in ILC subsets. (a) Analysis of lineage-specific transcription factor binding motifs enriched in completely demethylated promoters (CDMPs) and hydroxymethylated promoters (HMPs). The T-bet, GATA3 and ROR γ t binding motifs were identified from ChIPBase v3.0 database. The normalized tag density of T-bet, Gata3 and ROR γ t binding motifs within promoter regions were analyzed by using BEDTools and HOMER. Venn diagrams show overlapping CDMPs and HMPs between ILCP and ILC1 (left panel), ILC2 (middle panel), or ILC3 (right panel) comparison. (b) Analysis of lineage-specific transcription factor binding motifs enriched in completely

demethylated enhancers (CDMEs) and hydroxymethylated enhancers (HMEs). The T-bet, GATA3 and ROR γ t binding motifs were identified from ChIPBase v3.0 database. The normalized tag density of T-bet, Gata3 and ROR γ t binding motifs within enhancer regions were analyzed by using BEDTools and HOMER. Venn diagrams show overlapping CDMEs and HMEs between ILCP and ILC1 (left panel), ILC2 (middle panel), or ILC3 (right panel) comparison. (c-e) Analysis of methylated CpG island, enhancer, gene body regions and DEGs in ILC1s from *Zbtb16-Cre* and *Tet1^{flox/flox};Zbtb16-Cre* mice. Venn diagrams show overlapping methylated CpG island (c), enhancer (d) and gene body (e) regions between *Tet1^{+/+}* and *Tet1^{-/-}* ILC1s comparison (upper panel). The number of differentially expressed genes whose expression was correlated with differentially methylated CpG island (c), enhancer (d) and gene body (e) regions (the non-overlapping regions) were shown (lower panel). (f-h) Analysis of methylated CpG island, enhancer, gene body regions and DEGs in ILC2s and ILC3s from *Zbtb16-Cre* and *Tet1^{flox/flox};Zbtb16-Cre* mice. ILC2s and ILC3s from lamina propria of small intestine were isolated from *Zbtb16-Cre* and *Tet1^{flox/flox};Zbtb16-Cre* mice followed by mixing in 1:1 ratio for MeDIP-seq and RNA-seq. Venn diagrams show overlapping methylated CpG island (f), enhancer (g) and gene body (h) regions between *Tet1^{+/+}* and *Tet1^{-/-}* ILC2/3s comparison (upper panel). The number of DEGs whose expression was correlated with differentially methylated CpG island (f), enhancer (g) and gene body (h) regions from the non-overlapping regions in Venn diagrams were shown (lower panel).

Attached Fig. 4. Methylation regulation of gene elements in ILC1 under SPF and germ-free conditions. (a) Bar plot shows the percentage of DEGs that are correlated with differentially methylated and hydroxymethylated promoter, enhancer and gene body regions in ILC1s from mice under germ-free (GF) vs. SPF condition. ILC1s from lamina propria of small intestine were isolated from WT mice under SPF or GF condition followed by MeDIP-seq and hMeDIP-seq. The transcriptome data of ILC1s of WT mice under SPF and GF condition were from the GEO database (GSE85152). (b-d) Sankey diagram shows gene expression pattern correlated with differentially methylated and hydroxymethylated promoter (b), enhancer (c) and gene body (d) regions in ILC1s from mice under GF vs.

SPF condition. (e) TGF- β suppressed ILC1 differentiation from ILCPs *in vivo*. WT mice (2 weeks old) were treated with or without 100 ng TGF- β and/or 200 μ g TGF- β R1 inhibitor (SB431542) by intraperitoneal injection twice a week for 2 weeks. PBS treatment served as control (Ctrl) group. The percentage and cell number of ILC1s was analyzed by flow cytometry and shown as the mean \pm SEM. **, $p < 0.01$; ***, $p < 0.001$ by one-way ANOVA. $n = 4$ for each group. (f) Cholic acid suppressed ILC1 differentiation via TGR5 signaling *in vivo*. WT mice (2 weeks old) were orally gavaged with or without cholic acid (30 mg/kg) and/or TGR5 inhibitor (SBI-115, 15 mg/kg) for 14 consecutive days. PBS treatment served as control (Ctrl) group. The percentage and cell number of ILC1s was analyzed by flow cytometry and shown as the mean \pm SEM. *, $p < 0.05$ ***, $p < 0.001$ by one-way ANOVA. $n = 3$ for each group.

Attached Fig. 5. TET1 does not affect the effector gene expression and function in ILC2s and ILC3s. (a) Analysis of the indicated genes expression in ILC2s and ILC3s from *Zbtb16-Cre* and *Tet1^{fllox/fllox};Zbtb16-Cre* mice. ILC2s and ILC3s from lamina propria of small intestine were isolated from *Zbtb16-Cre* and *Tet1^{fllox/fllox};Zbtb16-Cre* mice and 1:1 mixed followed by RNA-seq. Gene expression of the indicated cytokines was shown in bar plot. (b-c) Gene set enrichment analysis (GSEA) of enriched genes in ILC2s and ILC3s from *Tet1^{fllox/fllox};Zbtb16-Cre* mice compared with *Zbtb16-Cre* mice. Gene sets of Th1 and Th2 cell differentiation (c) and Th17 cell differentiation (d) from the KEGG database (MSigDB) were used.

References

1. Bender MJ *et al.* Dietary tryptophan metabolite released by intratumoral *Lactobacillus reuteri* facilitates immune checkpoint inhibitor treatment. *Cell* (2023).
2. Chang, Y.L. *et al.* A screen of Crohn's disease-associated microbial metabolites identifies ascorbate as a novel metabolic inhibitor of activated human T cells. *Mucosal Immunol* **12**, 457-467 (2019).
3. Zelante, T. *et al.* Tryptophan Catabolites from Microbiota Engage Aryl Hydrocarbon Receptor and Balance Mucosal Reactivity via Interleukin-22. *Immunity* **39**, 372-385 (2013).
4. Andersson, R. *et al.* An atlas of active enhancers across human cell types and tissues. *Nature* **507**, 455-461 (2014).
5. Cosmas D. Arnold, D.G., Christoph Stelzer, Łukasz M. Boryn', Martina Rath, Alexander Stark. Genome-Wide Quantitative Enhancer Activity Maps Identified by STARR-seq. *science* **339**, 1074-1077 (2013).
6. Maqbool, M.A. *et al.* Alternative Enhancer Usage and Targeted Polycomb Marking Hallmark Promoter Choice during T Cell Differentiation. *Cell Reports* **32** (2020).
7. Corces, M.R. *et al.* The chromatin accessibility landscape of primary human cancers. *Science* **362** (2018).
8. Clark, S.J. *et al.* scNMT-seq enables joint profiling of chromatin accessibility DNA methylation and transcription in single cells. *Nature Communications* **9** (2018).

REVIEWERS' COMMENTS

Reviewer #1 (Remarks to the Author):

The authors have answered my questions and applied my suggestions. I have no more comments.

Reviewer #3 (Remarks to the Author):

The authors have addressed my concerns

Point-by-point response to the reviewers

Reviewer #1:

The authors have answered my questions and applied my suggestions. I have no more comments.

Answer: We appreciate the comments from this reviewer.

Reviewer #3:

The authors have addressed my concerns.

Answer: We appreciate the comments from this reviewer.